# An Optimized Franz-Parisi Criterion and its Equivalence with SQ Lower Bounds

**Siyu Chen**
Department of Statistics and Data Science
Yale University
siyu.chen.sc3226@yale.edu

**Theodor Misiakiewicz**
Department of Statistics and Data Science
Yale University
theodor.misiakiewicz@yale.edu

**Ilias Zadik**
Department of Statistics and Data Science
Yale University
ilias.zadik@yale.edu

**Peiyuan Zhang**
Department of Statistics and Data Science
Yale University
peiyuan.zhang@yale.edu

## Abstract

Bandeira et al. (2022) introduced the Franz-Parisi (FP) criterion for characterizing the computational hard phases in statistical detection problems. The FP criterion, based on an annealed version of the celebrated Franz-Parisi potential from statistical physics, was shown to be equivalent to low-degree polynomial (LDP) lower bounds for Gaussian additive models, thereby connecting two distinct approaches to understanding the computational hardness in statistical inference. In this paper, we propose a refined FP criterion that aims to better capture the geometric "overlap" structure of statistical models. Our main result establishes that this optimized FP criterion is equivalent to Statistical Query (SQ) lower bounds—another foundational framework in computational complexity of statistical inference. Crucially, this equivalence holds under a mild, verifiable assumption satisfied by a broad class of statistical models, including Gaussian additive models, planted sparse models, as well as non-Gaussian component analysis (NGCA), single-index (SI) models, and convex truncation detection settings. For instance, in the case of convex truncation tasks, the assumption is equivalent with the Gaussian correlation inequality (Royen, 2014) from convex geometry. In addition to the above, our equivalence not only unifies and simplifies the derivation of several known SQ lower bounds—such as for the NGCA model (Diakonikolas et al., 2017) and the SI model (Damian et al., 2024)—but also yields new SQ lower bounds of independent interest, including for the computational gaps in mixed sparse linear regression (Arpino et al., 2023) and convex truncation (De et al., 2023).

## 1 Introduction

Over the past decades, a central focus in statistical inference has been to understand the transition from computationally easy to hard regimes—that is, to characterize when a statistical task can be solved by polynomial-time algorithms. A key insight from this line of work is the emergence of *computational-statistical tradeoffs*: in many models, there exist broad parameter regimes where information-theoretic recovery is possible, yet no known polynomial-time algorithm succeeds.

39th Conference on Neural Information Processing Systems (NeurIPS 2025).

Evidence for such tradeoffs spans multiple disciplines with varying levels of mathematical rigor. In particular, the statistical physics community has played an instrumental role by leveraging non-rigorous but highly predictive techniques to study average-case hardness. Their approach typically analyzes the geometry of solution spaces and identifies structural properties that correlate with algorithmic intractability (see [40] for a survey). Remarkably, for many statistical models, the predictions from statistical physics have been in striking agreement with the performance of the best-known polynomial-time algorithms.

Alongside these heuristic predictions, rigorous frameworks from statistics and theoretical computer science have been developed to analyze the limitations of efficient algorithms. While ruling out *all* polynomial-time algorithms would require resolving $\mathcal{P} \neq \mathcal{NP}$, substantial progress has been made by studying broad, expressive classes of polynomial-time algorithms. Two frameworks have emerged as particularly influential: *low-degree* (LD) *polynomial* lower bounds [31] and *statistical query* (SQ) lower bounds [20]. For many "nice enough" detection problems, the lower bounds derived from these frameworks align closely with the performance of the best-known polynomial-time algorithms[1]. This striking consistency has motivated the formulation of the so-called *low-degree conjecture* by Hopkins [26], which posits that for sufficiently "symmetric and noisy" models, the failure of degree-$O(\log n)$ polynomials is indicative of the failure of all polynomial-time algorithms [31].

Given this context, a natural question arises: can one formally connect these two seemingly distinct approaches? At first glance, the answer appears negative, due to a fundamental mismatch in scope. Statistical physics techniques are primarily geared toward estimation problems, where the goal is to recover a hidden signal, while the rigorous frameworks discussed above—such as LD and SQ lower bounds—are focused on detection or hypothesis testing, where the task is to distinguish between the presence or absence of a signal in a noisy environment[2]. Nevertheless, a major step towards bridging this gap was taken by Bandeira et al. (2022) [6], who introduced the *Franz-Parisi* (FP) *criterion* for computational hardness in detection tasks. Inspired by the seminal work of Franz and Parisi in spin glass theory [21], the FP criterion provides a geometric perspective on computational hardness rooted in overlap structures. Crucially, Bandeira et al. showed that for Gaussian additive models, the FP criterion is mathematically equivalent to the low-degree (LD) lower bounds, thereby establishing a rigorous link between statistical physics heuristics and formal algorithmic barriers.

Specifically, consider the following general detection problem between two distributions $\mathbb{P}$ and $\mathbb{Q}$ supported on a subset of $\mathbb{R}^n$, which in what follows we refer to as **a "$\mathbb{P}$ versus $\mathbb{Q}$" task**. Under the *planted* distribution $\mathbb{P} = \mathbb{E}_u \mathbb{P}_u$, a signal $u$ is drawn from a prior distribution $\pi$ supported on $\Theta \subseteq \mathcal{S}^{N-1}$, and one observes $m$ independent samples $Y_1, \ldots, Y_m \sim \mathbb{P}_u$. Under the *null* distribution $\mathbb{Q}$, the samples are drawn independently from $Y_1, \ldots, Y_m \sim \mathbb{Q}$. The goal in the detection task[3] is to distinguish between these two hypotheses based on the observed data, that is to find a test statistics with vanishing Type I and Type II errors, as $n$ grows. Note that the computational question then is whether such a successful test statistic exists that also terminates in polynomial-in-$mn$ time. To characterize the hardness of detection problems from the statistical physics perspective, Bandeira et al. in [6] introduced the following notion of Franz-Parisi (FP) hardness:

**Definition 1** (FP hardness). *For $D, m \in \mathbb{N}, \varepsilon > 0$, we say that a $\mathbb{P}$ versus $\mathbb{Q}$ detection task is $(q, m, \varepsilon)$-FP hard if*

$$\textbf{FP:} \quad \mathbb{E}\left[\langle L_u^{\otimes m}, L_v^{\otimes m} \rangle \cdot \mathbf{1}(|\langle u, v \rangle| \leq \delta(q))\right] \leq 1 + \varepsilon, \quad \text{where} \tag{1}$$

$$\delta(q) = \sup\{\delta > 0 : \pi^2(|\langle u, v \rangle| \geq \delta) \geq q^{-2}\}. \tag{2}$$

In the definition we denoted as customary $L_u = \frac{d\mathbb{P}_u}{d\mathbb{Q}}, u \in \Theta$, and for $f, g \in \mathcal{L}^2(\mathbb{R}^n)$, the Hilbert space $L^2(\mathbb{Q})$ of (square integrable) functions from $\mathbb{R}^n$ to $\mathbb{R}$, we use $\langle f, g \rangle_{\mathbb{Q}} = \mathbb{E}_{Y \sim \mathbb{Q}} f(Y) g(Y)$.

---

[1] For exceptions to this correspondence, see [39] and discussion therein.

[2] Some recent work has extended low-degree lower bounds to estimation settings, beginning with [38], though this direction remains relatively underdeveloped.

[3] The associated *estimation* problem consists in recovering the planted signal $u$ from $Y_1, \ldots, Y_m \sim \mathbb{P}_u$.

We elaborate in Section A.1 on the statistical physics motivations behind this criterion, and only briefly highlight its core intuition here. The left-hand side of the FP condition integrates the function $F_{\mathrm{ann}}(t) := \mathbb{E}\left[\langle L_u^{\otimes m}, L_v^{\otimes m}\rangle \cdot \mathbf{1}(\langle u, v\rangle = t)\right]$, over a $(1 - q^{-2})$-typical region of the overlap variable $t$, corresponding to the constraint $|\langle u, v\rangle| \leq \delta(q)$. This function $F_{\mathrm{ann}}(t)$ is an annealed proxy for the Franz-Parisi potential, a central object in statistical physics that has long served as a predictor of algorithmic hardness [40]. Intuitively, the Franz-Parisi potential captures the energy landscape experienced by local algorithms—such as Langevin or Glauber dynamics—whose performance is constrained by the geometry of the underlying signal space. The overlap $\langle u, v\rangle$ naturally quantifies a local "geometric" similarity between signals, making it a meaningful argument for $F_{\mathrm{ann}}(t)$ and explaining its role within the FP criterion.

Returning to the definition of FP hardness, the parameter $m$ corresponds to the sample size, and one should interpret $q$ as a proxy for the required runtime. In this light, Bandeira et al. (2022) proved that, for Gaussian additive models, FP-hardness is equivalent to the failure of degree-$D = \log q$ polynomials to solve the detection task with $m$ samples—i.e., roughly the authors of [6] showed that the problem is $(q, m, O(1))$-FP hard if and only if it is "hard" for degree-$\log q$ polynomials to solve the detection task[4]. Hence, based on the current belief in the literature of low-degree lower bounds that a $D$-degree lower bound implies that the detection task requires at least $e^D$ runtime to be solved, e.g., see [18], proving a task is $((mn)^{\omega(1)}, m, O(1))$-FP hard for a Gaussian additive model provides rigorous evidence for polynomial-time hardness for the task.

Despite this success, the connection between the FP potential and other rigorous notions of algorithmic hardness remains limited. [6] only established a formal equivalence for Gaussian additive models and an one-sided implication for planted sparse models between the FP criterion and "low-degree" lower bounds. They further presented counterexamples where the equivalence fails entirely. In this work, our aim is to extend the Franz-Parisi criterion to rigorously characterize hardness beyond Gaussian additive models, and to clarify the scope and limitations of this framework across a broader class of statistical models.

## 1.1 Main Contributions

Our main contribution is to propose a slight modification of the FP-hardness criterion from [6], motivated by the observation that sticking to the Euclidean geometry assumption (and hence the "overlap" $\langle u, v\rangle$) may fail to capture the "true" hardness of some statistical models. We remark that this is an arguably natural modification, as (1) there are many statistical models for which the Euclidean geometry appears unnatural for navigating their parameter space (see Section 5 for a simple such construction), and (2) even in statistical physics settings, the Franz-Parisi potential is often considered under a more general notion of overlap [22]. Motivated by these considerations, we propose optimizing the "overlap" event inside the FP-hardness definition, subject only to a mild symmetry assumption for technical reasons. This leads to the following new criterion of FP-hardness:

**Definition 2** (Generalized Franz Parisi (GFP) hardness under symmetry $G$). *Fix $q, m \in \mathbb{N}, \varepsilon > 0$ and a group $G$ of finite order acting on the parameter space of the signal. We say a "$\mathbb{P}$ versus $\mathbb{Q}$" problem is $(q, m, \varepsilon)$-GFP$_G$ hard if*

$$\textbf{GFP}_G: \quad \inf_{\substack{A : \pi^{\otimes 2}(A) \geq 1 - q^{-2} \\ A \text{ is } G^2\text{-invariant}}} \mathbb{E}\left[\langle L_u^{\otimes m}, L_v^{\otimes m}\rangle_{\mathbb{Q}} \mathbf{1}(A)\right] \leq 1 + \varepsilon. \tag{3}$$

As in the original FP-hardness framework of [6], one should interpret $q$ as a proxy for runtime, and therefore $((mn)^{\omega(1)}, m, O(1))$-GFP hardness should be providing evidence of polynomial-time hardness with $m$ samples in this framework. We highlight that the assumption on the invariance of the optimizing event under group $G$ is made for technical reasons to enhance the applicability of our hardness criterion. We point the reader to Section 3.1 for further discussion of this assumption.

The main result of this work is that the "optimized" notion of GFP-hardness is fundamentally connected with the well-established framework of Statistical Query (SQ) hardness. The SQ framework

---

[4][6] established this equivalence for $m = 1$, but the argument extends directly to general $m$.

was initially proposed by Kearns in [30] to capture the power of noise-tolerant algorithms. The notion of a statistical dimension proposed by [20] allowed for achieving powerful lower bounds against SQ methods, which we refer to from now on as SQ-hardness results. We employ here a slight strengthening of the notion of SQ-hardness from [20], introduced in [8].

**Definition 3** (SQ hardness). *Fix $q, m \in \mathbb{N}$. We say a "$\mathbb{P}$ versus $\mathbb{Q}$" detection problem is $(q, m)$-SQ hard if*

$$\text{SQ:} \quad \sup_{A:\pi^2(A) \geq q^{-2}} \mathbb{E}\left[\left|\langle L_u, L_v \rangle_{\mathbb{Q}} - 1\right| \mid A\right] \leq \frac{1}{m}. \tag{4}$$

Roughly, a detection problem is $(q, m)$-SQ hard if any Statistical Query method succeeding at solving it with $m$ samples requires $q$ queries, which should be interpreted as requiring runtime $q$ (see [8, Appendix A] for more details and motivation). Hence, proving a task is $((mn)^{\omega(1)}, m)$-SQ hard provides evidence for polynomial-time hardness for the task.

Our main result is informally described as follows.

**Theorem 1.** *(Informal, GFP and SQ equivalence) Consider any $\mathbb{P}$ versus $\mathbb{Q}$ detection task which we assume (1) it satisfies a mild assumption with respect to a group $G$ of finite order acting on the parameter space (namely Assumption 1 below), and (2) it is information-theoretically impossible to be solved with $m_{\mathrm{IT}}$ samples. Then the following holds for any samples size $m$ and proxy runtime $q = m^{\Omega(1)}$.*

- *If the task is $(q, m)$-SQ hard, then it is also $(\Theta(q), \Theta(m), O(1))$-GFP$_G$-hard.*

- *If the task is $(q, m, O(1))$-GFP$_G$-hard, then it is also $(m^{\Theta(m_{\mathrm{IT}})}, m^{1-o(1)})$-SQ hard.*

Note that often in statistical tasks of interest $m_{\mathrm{IT}} = \omega(\log n)$ (in fact, more often than not $m_{\mathrm{IT}} = \mathrm{poly}(n)$). Under this condition, Theorem 1 implies that a task is $((mn)^{\omega(1)}, m^{1-o(1)}, O(1))$-GFP hard if and only if it is $((mn)^{\omega(1)}, m^{1-o(1)})$-SQ hardness, matching the two criteria for hardness.

On top of that, as we mentioned above and discuss in Section 3.1, the required Assumption 1 on the detection task is rather mild. In fact, it turns out that it is satisfied for several models of recent interest in the community, making a strong case of how the Generalized Franz-Parisi criterion now correctly predicts the hardness phase for them. Importantly, these models include the Gaussian additive models and also greatly extend beyond them, significantly extending the key message from [6] about connecting the physics-based forms of hardness to more rigorous frameworks. We list now some of the tasks that satisfy Assumption 1.

1. *All Gaussian additive models* (GAMs), under any symmetric prior, satisfy Assumption 1 with $G = \mathbb{Z}_2$ that flips the sign of the signal. Moreover, in that case the Generalized Franz Parisi criterion is equivalent to the Franz-Parisi criterion, that is the optimizing event $A$ in (3) is of the form $\{|\langle u, v \rangle| \leq \delta(q)\}$. Hence, Theorem 1 allows us to extend the result of [6] which proved the equivalence of FP-hardness to Low-degree hardness for GAMs, to also proving FP-hardness equivalent with SQ-hardness for these settings[5].

2. *All Planted Sparse Models* satisfy Assumption 1 for the trivial group $G = \{\mathrm{id}\}$. In particular, using Theorem 1 we can prove that GFP-hardness is equivalent to SQ-hardness for multiple well-studied models such as sparse phase retrieval [5], sparse regression [24, 6], (multi-sample) sparse PCA [7], and Bernoulli group testing [11]. As a corollary of this connection, we present a straightforward argument to obtain an SQ lower bound for the mixed sparse linear regression problem [5]. We remark that in [6] it has been proven that FP-hardness implies low-degree hardness for all Planted Sparse Models, but no result was presented for the other direction.

3. *All Non-Gaussian component analysis (NGCA) models* and *all single-index models (under any symmetric prior)* satisfy Assumption 1 with $G = \mathbb{Z}_2$, Therefore, via Theorem 1 GFP-hardness is again equivalent with SQ-hardness for these tasks.

---

[5]We remark that such a connection could also be made via the results of [8], since GAMs are noise-robust.

4. *All Gaussian convex truncation models* satisfy Assumption 1 for $G = \{\text{id}\}$. In particular, interestingly Assumption 1 for these models is exactly equivalent to the celebrated Gaussian correlation inequality for convex bodies in probability theory, which was a multi-decade open problem posed in 1972 in [25] that was finally proven by Royen in 2014 [37]. Leveraging the equivalence between GFP-hardness and SQ-hardness in Theorem 1, we establish an SQ-lower bound for the convex truncation detection task. This allows us to provide, to the best of our knowledge, the first formal evidence that the current state-of-the-art polynomial-time detection method for convex truncation proposed in [15] has optimal sample complexity.

We also complement our results, with a simple example satisfying Assumption 1 where FP-hardness does *not* coincide with GFP-hardness, which we interpret as a model where the Euclidean geometry is not appropriate. We finally conclude the paper with a discussion.

For completeness, we prove in Appendix B the equivalence between GFP-hardness and low-degree (LD) polynomial hardness for noise-robust models. This result follows by combining our GFP–SQ equivalence with the equivalence between SQ-hardness and LD-hardness under noise robustness shown by Brennan et al. [8]. In particular, our GFP-hardness results for the examples presented in this paper immediately imply low-degree lower bounds in all those settings. This substantially extends the equivalence established in [6]. For clarity and readers' convenience, we also include succinct proofs of the SQ-to-LD equivalence, adapted from [8].

## 2 Setting and Definitions

We first recall the definition of **a "$\mathbb{P}$ versus $\mathbb{Q}$" task** mentioned in the Introduction. Under the *planted* distribution $\mathbb{P} = \mathbb{E}_u \mathbb{P}_u$, a signal $u$ is drawn from a prior distribution $\pi$ supported on $\Theta \subseteq \mathcal{S}^{N-1}$, and one observes $m$ independent samples $Y_1, \ldots, Y_m \sim \mathbb{P}_u$. Under the *null* distribution $\mathbb{Q}$, the samples are drawn independently from $Y_1, \ldots, Y_m \sim \mathbb{Q}$. The goal in the detection task[6] is the so-called strong detection task to distinguish between these two hypotheses based on the observed data, that is to find a test statistics with vanishing Type I and Type II errors, as $n$ grows. We will also be interested in the weak detection task, which is that the sum of type I and type II errors is at most $1 - \varepsilon$ for some fixed $\varepsilon > 0$ (not depending on $n$). In other words, strong detection means the test succeeds with high probability, while weak detection means the test has some non-trivial advantage over random guessing.

Throughout, we will work in the Hilbert space $L^2(\mathbb{Q})$ of (square integrable) functions $\mathbb{R}^N \to \mathbb{R}$ with inner product $\langle f, g \rangle_{\mathbb{Q}} := \mathbb{E}_{Y \sim \mathbb{Q}}[f(Y)g(Y)]$ and corresponding norm $\|f\|_{\mathbb{Q}} := \langle f, f \rangle_{\mathbb{Q}}^{1/2}$. We will assume that $\mathbb{P}_u$ is absolutely continuous with respect to $\mathbb{Q}$ for all $u \in \text{supp}(\pi)$, use $L_u := \frac{d\mathbb{P}_u}{d\mathbb{Q}}$ to denote the likelihood ratio, and assume that $L_u \in L^2(\mathbb{Q})$ for all $u \in \text{supp}(\pi)$. The likelihood ratio between $\mathbb{P}$ and $\mathbb{Q}$ is denoted by $L := \frac{d\mathbb{P}}{d\mathbb{Q}} = \mathbb{E}_{u \sim \mu} L_u$. Observe that for $m$ samples, we denote by $L_m = \mathbb{E}_{u \sim \mu} L_u$ the $m$-sample likelihood ratio. Finally, for a function $f : \mathbb{R}^N \to \mathbb{R}$ and integer $D \in \mathbb{N}$, we let $f^{\leq D}$ denote the orthogonal (w.r.t. $\langle \cdot, \cdot \rangle_{\mathbb{Q}}$) projection of $f$ onto the subspace of polynomials of degree at most $D$.

An important identity between the (squared) norm of the likelihood ratio with $m$ samples and the *chi-squared divergence* $\chi^2(\mathbb{P}^{\otimes m} \| \mathbb{Q}^{\otimes m})$ is

$$\|L\|_{\mathbb{Q}}^2 = \|\mathbb{E}_{u \sim \mu} L_u\|_{\mathbb{Q}}^2 = \chi^2(\mathbb{P} \| \mathbb{Q}) + 1 \geq 1 \,.$$

This quantity has the following standard implications for *information-theoretic* impossibility of testing, in the asymptotic regime $n \to \infty$. The proofs can be found in e.g. [34, Lemma 2].

- If $\|L\|_{\mathbb{Q}}^2 = O(1)$ then strong detection is impossible.
- If $\|L\|_{\mathbb{Q}}^2 = 1 + o(1)$ then weak detection is impossible.

---

[6]The associated *estimation* problem consists in recovering the planted signal $u$ from $Y_1, \ldots, Y_m \sim \mathbb{P}_u$.

# 3 Main Results

In this section, we formally present our equivalence between GFP-hardness and SQ-hardness.

## 3.1 The Assumption

As mentioned in the Introduction, all our results operate under a crucial assumption on the "$\mathbb{P}$ versus $\mathbb{Q}$" detection task. The assumption is as follows.

**Assumption 1.** *Given any "$\mathbb{P}$ versus $\mathbb{Q}$" task, there exists a $\pi$-preserving finite group $G$ acting on the parameter space $\Theta$, i.e., for all $g \in G$, $g(v) \overset{(d)}{=} v$ for $v \sim \pi$, such that for any sample size $m$ for any $u, v \in \Theta$, the following "correlation" inequality holds for any $k \in \mathbb{N}$*

$$\mathbb{E}_{g,g' \sim Unif(G)}(\langle L_{g(u)}, L_{g'(v)} \rangle_{\mathbb{Q}} - 1)^k \geq 0. \tag{5}$$

We first remark that (5) is a natural condition even if $G$ is the trivial group, $G = \{\mathrm{id}\}$. Indeed in that case (5) asks that for all $u, v \in \Theta$,

$$\langle L_u, L_v \rangle_{\mathbb{Q}} \geq 1. \tag{6}$$

Recall that if one averages over all $(u, v) \sim \pi^{\otimes 2}$, we have by standard identities

$$\mathbb{E}\langle L_u, L_v \rangle_{\mathbb{Q}} = \mathbb{E}_{\mathbb{Q}}\|\mathbb{E}_u L_u\|_2^2 = 1 + \chi^2(\mathbb{P}, \mathbb{Q}) \geq 1.$$

Thus, (6) should be understood as a pointwise condition that is guaranteed to hold in expectation over the product measure $\pi^{\otimes 2}$ for any $\mathbb{P}, \mathbb{Q}$. While this pointwise condition turns out to be vanilla satisfied in many models (such as Planted Sparse Models or Convex Truncation settings), a slight modification of it—leading to (5)—applies more broadly. Specifically, this modified condition requires (6) to hold for a pair $u, v$ only after performing a "small" averaging over the a group orbit that preserves the prior $\pi$. For instance, if the prior is symmetric around 0 and the group $G$ is $\mathbb{Z}_2$, which acts by flipping the sign of the signal $u$, then for $k = 1$, condition (5) reduces to demonstrating that, for all $u, v$,

$$\frac{1}{4}(\mathbb{E}\langle L_u, L_v \rangle_{\mathbb{Q}} + \mathbb{E}\langle L_{-u}, L_v \rangle_{\mathbb{Q}} + \mathbb{E}\langle L_u, L_{-v} \rangle_{\mathbb{Q}} + \mathbb{E}\langle L_{-u}, L_{-v} \rangle_{\mathbb{Q}}) \geq 1,$$

which is significantly less restrictive than the original pointwise condition (6). This averaging approach allows for much greater generality, making it applicable to various settings, including Gaussian additive models, single-index models, and Non-Gaussian component analysis settings.

**Remark 3.1.** We finally make a trivial remark that will be useful in verifying (5) in our examples in Section 4 with symmetric prior. In all of them by symmetry we have for all $u, v$ $\langle L_u, L_{-v} \rangle_{\mathbb{Q}} = \langle L_{-u}, L_v \rangle_{\mathbb{Q}}$ and $\langle L_u, L_v \rangle_{\mathbb{Q}} = \langle L_{-u}, L_{-v} \rangle_{\mathbb{Q}}$. Using that and the trivial fact that for all $x, y \in \mathbb{R}$, if $x + y \geq 0$ then $x^k + y^k \geq 0$ for all $k \in \mathbb{N}$, we conclude that if $G$ is either the trivial group or $\mathbb{Z}_2$ (which will be the case in all examples of Section 4) it suffices to check the case $k = 1$ in (5), and then it automatically holds for all $k \in \mathbb{N}$.

**Remark 3.2.** As mentioned in the previous remark, we highlight that in all our examples in Section 4 of our GFP-SQ equivalence theorem below, we either use $G$ to be the trivial group or $\mathbb{Z}_2$. The reason we state our assumption Assumption 1 for a general finite group $G$ is for potential further applications of our work.

## 3.2 The GFP-SQ equivalence

### 3.2.1 Simplifying GFP-hardness

We present our equivalence theorem in two steps. First, we identify an approximate optimal "overlap" event $A$ in the definition of GFP-hardness, which simplifies GFP-hardness significantly and makes GFP-hardness easier to establish in applications. Then, we prove the equivalence between this simplified version and SQ-hardness.

Given a group $G$ acting on the parameter space of the signal, it turns out that the approximately optimal "overlap" event $A$ takes the form $\{\rho_G(u,v) \leq r\}$ for the following notion of "overlap" between $u, v$,

$$\rho_G(u,v) = \max_{g,g' \in G} \{|\langle L_{g(u)}, L_{g'(v)} \rangle_{\mathbb{Q}} - 1|\}.$$

In particular, focusing only on such type of events we define the following version of FP-hardness.

**Definition 4** ($\rho_G$-FP hardness). *Fix $q, m \in \mathbb{N}$, $\varepsilon > 0$ and a finite group $G$ acting on the parameter space of the signal. We say a "$\mathbb{P}$ versus $\mathbb{Q}$" problem is $(q, m, \varepsilon)$-$\rho_G$-FP hard if*

$$\rho_G\text{-}\textbf{FP}: \quad \mathbb{E}\left[\langle L_u^{\otimes m}, L_v^{\otimes m} \rangle_{\mathbb{Q}} \cdot \mathbf{1}(\rho_G(u,v) < r(q))\right] \leq 1 + \varepsilon, \quad \text{where} \tag{7}$$

$$r(q) = \sup\{r : \pi^2(\rho_G(u,v) \geq r) \geq q^{-2}\}, \tag{8}$$

We prove that GFP$_G$-hardness is equivalent to $\rho_G$-FP hardness under Assumption 1.

**Theorem 2.** *Consider any "$\mathbb{P}$ versus $\mathbb{Q}$" task that satisfies Assumption 1 for a group $G$. Suppose $m, q \in \mathbb{N}$ and $\varepsilon > 0$. Then the following statements hold.*

1. *If the task is $(q, m, \varepsilon)$-$\rho_G$-FP hard, then the task is also $(q, m, \varepsilon)$-GFP$_G$ hard.*

2. *Assume there exists an $r > 0$ such that $\pi^2(\rho_G(u,v) < r) = 1 - q^{-2}$ and that $m$ is even. Then, if the task is $(q, m, \varepsilon)$-GFP$_G$ hard, then it is $(q, m, \frac{3}{|G|}(1+\varepsilon) + m \cdot \chi^2(\mathbb{P}, \mathbb{Q}))$-$\rho_G$-FP hard. In particular, if $m\chi^2(\mathbb{P}, \mathbb{Q}) = O(1)$, the task is $(q, m, O(1 + \varepsilon))$-$\rho_G$-FP hard.*

The proof of this theorem can be found in Appendix C.1.

**Remark 3.3.** While the first implication is immediate to grasp, the second implication has some additional conditions we now elaborate upon. First, both the requirements of the existence of $r$ with the desired probability mass and the parity of $m$ are for technical convenience, and both can be easily remove with some tedious work. Second, any potential "blow-up" in the $\varepsilon$-term for $\rho_G$-FP hard depends only on $|G|$, which should be treated as constant, and the term $m \cdot \chi^2(\mathbb{P}, \mathbb{Q})$, which is an easy to compute quantity (usually $n = 1$ and $\chi^2(\mathbb{P}, \mathbb{Q})$ is an one-dimensional integral). Moreover, it is almost always of order $O(1)$ for detection tasks that are conjecturally hard with $m$ samples. Indeed, the mathematical reason behind this is exactly that it is equal to the squared $\mathcal{L}^2$-norm of the projection of the likelihood onto the degree-1 polynomial space, i.e., on the span of linear functions. On top of that, if the detection task is $(q, m)$-SQ hard *for any* $q$ then it holds directly $m\chi^2(\mathbb{P}, \mathbb{Q}) = O(1)$ as well. We elaborate more on this in Remark B.1 in Section B.

### 3.2.2 The equivalence

As we have already proven an equivalence between GFP-hardness and $\rho_G$-FP hardness, it suffices to connect the latter with SQ-hardness. This is the topic of the next theorem.

**Theorem 3** (SQ and $\rho_G$-FP Equivalence). *Suppose a "$\mathbb{P}$ versus $\mathbb{Q}$" task satisfies Assumption 1 for a group $G$.*

1. *If the task is $(q, m)$-SQ hard for some $q, m$ with $q > 2$ then, it is also $(q', m', e|G|^{-1}m'/m)$-$\rho_G$-FP hard for any integers $q' < q/\sqrt{2}$ and $m' \leq m/2$.*

2. *Suppose the task is $(q, m, \varepsilon)$-$\rho_G$-FP hard for some $q, m$ integers. Assume that there exists an $r = r(q) > 0$ such that $\pi^2(\rho_G(u,v) < r) = 1 - q^{-2}$ and $m$ is even. Then, the model is also $(q', m')$-SQ hard for any even integer $t$ with $t \leq \log q/\log m$ and any integer $q' > 0$, where*

$$m' = \frac{m}{(t(1+\varepsilon)^{1/t} + \chi^2(\mathbb{P}^{\otimes 4t} \| \mathbb{Q}^{\otimes 4t}))(q')^{2/t}}.$$

*In particular, if for some sample size $m_{\mathrm{IT}}$, we have*

(a) *(Bounded $\chi^2$ for $m_{\mathrm{IT}}$ samples)*

$$\chi^2(\mathbb{P}^{\otimes m_{\mathrm{IT}}} \| \mathbb{Q}^{\otimes m_{\mathrm{IT}}}) = O(1).$$

*(b) (Large enough q)*

$$q \geq m^{m_{\mathrm{IT}}}$$

*then the model is $(m^{\delta m_{\mathrm{IT}}}, \Theta(\frac{m^{1-O(\delta)}}{m_{\mathrm{IT}}(1+\varepsilon)}))$-SQ hard for any $\delta > 0$.*

The proof of this theorem can be found in Appendix C.2. Similar to Theorem 2, the conditions on $r, m$ of part 2 in Theorem 3 are for technical convenience and can be easily removed. As we discussed in the Introduction the assumption that there exists some sufficiently growing $m_{\mathrm{IT}}$ (e.g., growing super-logarithmically in $n$) is natural for multiple commonly studied models. We remark that the condition on the information theory threshold $m_{\mathrm{IT}}$ to be growing with $n$ is also necessary, by constructing a variant of the planted clique problem which satisfies Assumption 1, it is not SQ-hard and is GFP-hard. Lastly, we also note that our introduced Assumption 1 is also necessary for the equivalence. In Section A, we discuss a counterexample not satisfying Assumption 1 that is GFP-hard, but not SQ-hard.

**Remark 3.4.** We note that while our bounds in the equivalence of Theorem 2 deteriorate when $|G|$ becomes large, a slightly more general equivalence between GFP and SQ, using a variant of $\rho_G$, can also be proven for infinite groups $G$ under an "hypercontractivity" assumption on $\langle L_u, L_v \rangle_{\mathbb{Q}}$ with respect to the pair $(u, v)$. We omit this generalization as in all relevant examples in this work a small group action using either the trivial or 2-cyclic group suffices.

## 4 Examples

In this section, we discuss two popular classes of detection tasks that satisfy Assumption 1 and hence fall under our GFP-SQ equivalence. Further examples are deferred to Appendix D.

### 4.1 Gaussian Additive Models

A $\mathbb{P}$ versus $\mathbb{Q}$ task is a Gaussian additive model (GAM) if it satisfies:

1. Under the null model, $\mathbb{Q} = \mathcal{N}(0, I_n)$.
2. Under the planted model $\mathbb{P}_u$ (for $u \in S^{n-1}$), for some signal-to-noise ratio (SNR) $\lambda > 0$ we set

$$Y = \lambda u + Z, \qquad \text{for some } Z \sim \mathbb{Q}.$$

GAMs includes multiple well-studied models in the literature, with the predominant examples being (multisample variants) of tensor PCA [36] and sparse PCA [2]. For such models, it can be straightforwardly checked (see [6, Proposition 2.3]) that for all $u, v$,

$$\langle L_u, L_v \rangle_{\mathbb{Q}} = e^{\lambda^2 \langle u, v \rangle}.$$

So for instance, in the case of non-negative sparse PCA where $u, v$ are binary $k$-sparse vectors in (see e.g., [4, 10]) we always have $\langle u, v \rangle \geq 0$, and therefore Assumption 1 is always satisfied for the trivial group $G$. On top of that, Assumption 1 remains true for any prior which is symmetric around 0; this time Assumption 1 is also always satisfied by choosing the action of $G = \mathbb{Z}_2$ which flips the sign of $u$. We remark that symmetric priors encompass most commonly used priors for GAMs, e.g., for tensor PCA where $u = \mathrm{vec}(x^{\otimes r}), x \sim \mathrm{Unif}(S^{d-1})$.

**Lemma 1.** *Consider any GAM with symmetric $\pi$, i.e., $v = -v, v \sim \pi$. For any $u, v \in \mathrm{support}(\pi)$,*

$$\frac{1}{4}(\langle L_u, L_v \rangle_{\mathbb{Q}} + \langle L_{-u}, L_v \rangle_{\mathbb{Q}} + \langle L_u, L_{-v} \rangle_{\mathbb{Q}} + \langle L_{-u}, L_{-v} \rangle_{\mathbb{Q}}) \geq 1.$$

*Moreover, any GAM satisfies Assumption 1 for $G = \mathbb{Z}_2$ acting by flipping the sign of $u$.*

*Proof.* Notice

$$\frac{1}{4}(\langle L_u, L_v \rangle_{\mathbb{Q}} + \langle L_{-u}, L_v \rangle_{\mathbb{Q}} + \langle L_u, L_{-v} \rangle_{\mathbb{Q}} + \langle L_{-u}, L_{-v} \rangle_{\mathbb{Q}}) = \frac{1}{2}(\exp\left(\lambda^2 \langle u, v \rangle\right) + \exp\left(-\lambda^2 \langle u, v \rangle\right)) \geq 1.$$

Hence, given Remark 3.1, the conclusion follows. $\qquad\qquad\square$

Given the above lemma, we conclude the (almost) equivalence between GFP-hardness and SQ-hardness from Theorem 3.

**Remark 4.1.** We remark that in the symmetric prior case for a GAM and $G = \mathbb{Z}_2$ acting by flipping the sign of $u$, $\rho_G(u, v) = \exp\left(\lambda^2 |\langle u, v \rangle|\right)$ is an increasing function of $|\langle u, v \rangle|$. Hence, for such GAMs we conclude via Theorem 2 that FP-hardness is equivalent to GFP-hardness, and therefore also to SQ-hardness. This is in agreement with the results of [6] establishing that FP-hardness is equivalent to LD-hardness; in fact, our approach can offer an alternative proof of their result via the LD-SQ equivalence [8] and the noise robustness of GAMs (see Theorem Theorem 6).

## 4.2 Planted Sparse Models

In [6], the authors introduced the family of planted sparse models (PSM) and proved that FP-hardness for a PSM implies it's also low-degree hard. We start with the definition.

A $\mathbb{P}$ versus $\mathbb{Q}$ task is a planted sparse model (PSM) if it satisfies:

1. Under the null model, the one sample is given by $Y = (Y_1, \ldots, Y_n) \sim \mathbb{Q}$, where each entry $Y_i, i = 1, \ldots, n$ is drawn independently from some distribution $\mathbb{Q}_i, i = 1, \ldots, n$ on $\mathbb{R}$.

2. Under the planted model $\mathbb{P}_u$, we associate $u$ with a set of planted entries $\Phi_u \subset [n]$. Then on sample is generated as follows. For the entries $i \notin \Phi_u$, we draw $Y_i$ independently from $Q_i$ (which is identical as in the $\mathbb{Q}$ measure). For the entries in $\Phi_u$ we draw from an arbitrary joint distribution $P_u|_{\Phi_u}$ with the following symmetry condition: for any subset $S \subseteq \Phi_u$, the marginal distribution $P_u|_{\phi_u}(S)$ does not depend on $u$ but only on $S$, i.e. $P_u|_S = P_S$.

Multiple well-known detection models satisfy this definition, such as, a well studied model of sparse regression [24, 6], Bernoulli group testing [1, 11], sparse phase retrieval [5], as well as multi-sample variants [8] of planted clique [29] and sparse (Wigner) PCA [2].

Satisfyingly, all planted sparse models directly satisfy Assumption 1 for the trivial group. In fact this result has already been established for a different use in [6, Proposition 3.6], proving that any $u, v$ we have $\langle L_u, L_v \rangle_{\mathbb{Q}} \geq 1$. We state here for completeness.

**Lemma 2.** *Consider any PSM. For any $u, v \in \text{support}(\pi)$, $\langle L_u, L_v \rangle_{\mathbb{Q}} \geq 1$. This is to say, any PSM satisfies Assumption 1 for the trivial group.*

The proof follows from [6, Proposition 3.6] for $D = 0$. Using this, one can apply our main equivalence Theorem 3 to multiple interesting planted sparse models and obtain old and new SQ-hardness results in a rather streamlined fashion. As an instantiation of this, in Appendix D.1 we prove the GFP-hardness for the mixed sparse linear regression setting studied in [5] in its conjecturally hard regime. We then use our equivalence theorem to translate it into an SQ-hardness result in the same regime. Our results complement the existing low-degree lower bound [5], providing further evidence for the hard phase of the problem.

### 4.2.1 Other examples

Due to space constraint, we defer the following examples to Appendix D:

- *Non-Gaussian Component Analysis (NGCA):* Assumption 1 holds with $G = \mathbb{Z}_2$ for any symmetric prior. We recover the SQ-hardness result of [16] for the uniform prior via its equivalence with GFP-hardness, and establish a new SQ lower bound under a sparse prior.

- *Single-Index Models (SIM):* Again, Assumption 1 holds with $G = \mathbb{Z}_2$. We rederive the SQ-hardness result of [12] for the uniform prior through the GFP-hardness equivalence, and prove a new SQ lower bound for sparse priors.

- *Convex truncation detection:* Here Assumption 1 holds with the trivial group. In fact this assumption is precisely equivalent to the celebrated Gaussian Correlation Inequality on convex bodies [37, 32]. Using the GFP-hardness correspondence, we derive a new SQ lower bound that matches the current state-of-the-art polynomial-time algorithm of [15].

# 5 GFP-hardness is not always equal to FP-hardness

Recall that by definition, FP-hardness implies GFP-hardness. In this section, we show that the converse does not necessarily hold: we construct a $\mathbb{P}$ versus $\mathbb{Q}$ detection task that satisfies Assumption 1 and is easy under the FP criterion but hard under the GFP criterion. In particular, by using Theorem 2 and Theorem 3, the problem is also SQ-hard. Thus, while the FP criterion fails to capture the SQ-hardness in this case, our optimized GFP criterion correctly predicts it. As our initial departure from FP-hardness was that in many models the Euclidean overlap $\langle u, v \rangle$ might not be the "correct" choice, our example is carefully creating a model where the natural "overlap" $\rho_G(u, v)$ (based on Theorem 3) is not a function of the Euclidean dot product.

The $\mathbb{P}$ versus $\mathbb{Q}$ problem is defined as follows. The null model is $\mathbb{Q} = \mathrm{Rad}\left(\frac{1}{2}\right)^{\otimes(n+1)}$, i.e., each coordinate is an independent Rademacher random variable. For a signal $u \in \{0,1\}^{n+1}$, the sample $x \sim \mathbb{P}_u$ is generated by drawing each coordinate independently according to

$$x_i = \begin{cases} +1, & \text{w.p. } \frac{1}{2} + r \cdot \frac{1-(1-\alpha)\cdot u_i}{2}, \\ -1, & \text{w.p. } \frac{1}{2} - r \cdot \frac{1-(1-\alpha)\cdot u_i}{2}, \end{cases} \tag{9}$$

where $\alpha, r \in (0, 1)$ are fixed constants to be chosen later. The following holds.

**Lemma 3.** *Let $u, v \in \{0,1\}^{n+1}$. For any $u, v \in \{0,1\}^{n+1}$, $\langle L_u, L_v \rangle_{\mathbb{Q}} = \prod_{i=0}^{n}\left(1 + r^2 \cdot \alpha^{u_i + v_i}\right)$.*

Notice that our construction importantly ensures that the likelihood ratio inner product $\langle L_u, L_v \rangle$ is not solely a function of $\langle u, v \rangle$, but instead has a more intricate dependence on $u$ and $v$. It is exactly this reason that leads to the discrepancy between GFP and FP hardness stated below.

**Theorem 4.** *There exist a two-point prior $\pi$ on $u$ such that, for $r = n^{-1/2}$, $\alpha = n^{-1+2\varepsilon}$, $m = n^{1-\varepsilon}$ and $D = n^{\varepsilon}$, where $\varepsilon > 0$ is any small constant, the following hold. The $m$-sample hypothesis testing problem $\mathbb{E}_{u \sim \pi}\mathbb{P}_u^{\otimes m}$ versus $\mathbb{Q}^{\otimes m}$ is $(e^{D/2}, m, \Theta(n^{-\varepsilon}))$-GFP hard but not $(n^{-1}, m, \exp(\Theta(n^{\varepsilon})))$-FP hard. Moreover, via our equivalence theorem the model is $(e^{n^{\Theta(\varepsilon)}}, n^{1-\Theta(\varepsilon)})$-SQ hard.*

The proof of this Theorem and the above Lemma can be found in Appendix E.

# 6 Conclusion

In this work, we generalize the Franz-Parisi (FP) criterion introduced by [6], motivated by the observation that the Euclidean dot product may not be the most natural geometry for all statistical task—a point partially illustrated by our example in Section 5. Our main result shows that optimizing the overlap event in the FP definition of [6] leads to a Generalized Franz-Parisi (GFP) hardness criterion, which is equivalent to SQ-hardness for models satisfying the mild Assumption 1. This assumption holds in a broad range of well-studied problems, including Gaussian additive models, planted sparse models, single-index models, and convex truncation. Our work significantly strengthens the theoretical foundation behind the (annealed) FP potential's predictions from statistical physics, but also opens several questions:

1. *(Algorithmic implications)* Does the optimal overlap function $\rho_G(u, v)$—as characterized in Theorem 2—yield meaningful algorithmic insights, particularly for local search or geometric methods?

2. *(The annealed potential)* Can similar equivalences be established for the original (also known as quenched) FP potential, or is the choice of the annealed version fundamental?

3. *(Interpretation of FP Area)* Why does the area under the FP curve appear to govern detection hardness? Is there some physical/algorithmic interpretation of this phenomenon?

4. *(Generalization to estimation)* Can our techniques be extended from detection to estimation tasks, for which the Franz-Parisi potential was originally introduced?

We believe these questions point toward promising future directions, with the potential to unify different approaches on the computational complexity of statistical inference.

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

# Contents

## A   Discussions on FP criterion and assumptions

In this Section, we provide additional discussions on the Franz-Parisi criterion and its connection with Statistical physics, as well as, on the necessity of our assumptions in our main theorem (Theorem 3).

### A.1   Connection of the FP criterion with statistical physics

We begin by discussing the connection between the Franz-Parisi (FP) criterion and statistical physics methods. For a more detailed overview and additional references, we refer the reader to [6, Section 1.3].

A natural algorithm for solving the estimation problem of recovering $u$ from $Y = (Y_1, \ldots, Y_m) \sim \mathbb{P}_u$ is to run some "local" dynamics (e.g., Langevin or Glauber dynamics) to sample from the posterior

$$\mathbb{P}(u|Y) \propto \pi(v)\mathbb{P}(Y|v) = \pi(v)\prod_{i=1}^{m}\mathbb{P}_v(Y_i), v \in \Theta,$$

where $Y = (Y_1, \ldots, Y_m)$. In statistical physics, a powerful heuristic exists for predicting the success of local dynamics in sampling from random distributions of the form $p_Y(v)\nu(v), v \in \Theta$ where $\nu$ is a reference measure and $Y \sim \mu$ is a "disorder". The heuristic approach is to check the monotonicity of the so-called Franz-Parisi potential defined as

$$F(t) = \mathbb{E}_{u\sim\nu, Y\sim\mu}\left[\log \mathbb{E}_{v\sim\nu}\left[p_Y(v)\mathbf{1}\left(d(v,u)=t\right)\right]\right], t \in [0,1],$$

where $d(\cdot,\cdot)$ is some notion of (normalized) distance between the states $u, v$ in agreement with the operations of the local dynamics on the state space. The prediction, introduced by Franz and Parisi in [21], is that local dynamics can efficiently sample from the distribution if and only if the potential is monotonic, i.e., it lacks "bad" local minima. Remarkably, this prediction has been empirically validated across a range of problems in statistical physics, often yielding accurate forecasts of algorithmic tractability. For instance, when $d$ is the Euclidean distance, this criterion has proven effective in the study of spin glasses [21]. Other, more intricate distance functions have also been used successfully in non-spin glass settings, such as binary fluids [22].

Now, returning to statistical estimation settings, researchers in statistical physics have applied this rule for $p_Y(v) := \mathbb{P}(Y|v) = \prod_{i=1}^{m} \mathbb{P}_v(Y_i)$ and $\nu := \pi$ to arrive at a prediction of success for "local" algorithms based on the geometry defined by the distance $d$. The prediction [40] is then based on the monotonicity of the curve

$$F(t) = \mathbb{E}_{u\sim\pi, Y\sim\mathbb{P}_u}\left[\log \mathbb{E}_{v\sim\pi}\left(\mathbb{P}(Y|v)\mathbf{1}\left[d(v,u)=t\right]\right)\right], t \in [0,1],$$

or equivalently for

$$F(t) = \mathbb{E}_{u\sim\pi, Y\sim\mathbb{P}_u}\left[\log \mathbb{E}_{v\sim\pi}\left(\prod_{i=1}^{m}\frac{\mathbb{P}_v(Y_i)}{\mathbb{Q}(Y_i)}\mathbf{1}\left(d(v,u)=t\right)\right)\right], t \in [0,1], \tag{10}$$

Interestingly, when $d(\cdot, \cdot)$ is the Euclidean distance, recent mathematical works have indeed produced one-sided results linking the potential to the performance of local methods for estimation tasks in the context of the so-called Gaussian additive models (e.g., [3, 4, 6]). This connection with the choice of the Euclidean distance can be perhaps cast as natural by a well-known analogy between spin glasses and GAMs, where GAMs often take the form of "spiked" spin glass models. Now, given the above successes, both heuristic and rigorous, it is natural to conjecture a potential link between general algorithmic hardness and the monotonicity of $F(t)$. However, this connection remains unproven in general, and known counterexamples exist. For instance, in sparse tensor PCA [10], there are regimes where the FP potential is non-monotonic (suggesting hardness), but some polynomial-time methods do succeed.

Despite the above issue, [6] used the Franz-Parisi potential to arrive at a different criterion, but now for algorithmic hardness of detection. Following an application of Jensen's inequality described in [6, Section 1.3] one get the following "annealed" upped bound for any $t \in [0, 1]$ $F(t) \leq \log \tilde{F}(t)$ for,

$$\tilde{F}(t) = \mathbb{E}_{u,v \sim p}, [\langle L_u^{\otimes m}, L_v^{\otimes m} \rangle 1 \, [d(v, u) = t]], t \in [0, 1]. \tag{11}$$

Then by focusing on the Euclidean distance $d$ (or equivalently the Euclidean dot product $\langle u, v \rangle$) they suggested the Franz-Parisi (FP) criterion Definition 1, restated here.

**Definition 5** (FP hardness). *We say a problem is $(q, m, \varepsilon)$-FP hard if*

$$\textbf{\textit{FP:}} \quad \mathbb{E}\left[\langle L_u^{\otimes m}, L_v^{\otimes m} \rangle \cdot \mathbf{1}(|\langle u, v \rangle| \leq \delta(q))\right] \leq 1 + \varepsilon, \quad \text{where} \tag{12}$$

$$\delta(q) = \sup\{\delta : \pi^2(\langle u, v \rangle \geq \delta) \geq q^{-2}\}, \tag{13}$$

Notice that the FP criterion says that to check for "hardness" of detection one should integrate the annealed FP curve is an $(1 - q^{-2})$-typical overlap $t$-region. Moreover, as we elaborated in the Introduction, one should understand $q$ in the above definition as a proxy for $q$ run-time. In that light, [6] roughly proved that for any GAMs is a $(q, m, O(1))$-FP hard if and only if $D = \log q$-degree polynomials fail to detect between $\mathbb{P}$ and $\mathbb{Q}$ with $m$ samples. We remark that, albeit this is an equivalence for detection, this is a first-of-a-kind result for GAMs as it is mathematical connection between the FP curve and a rigorous form of hardness. However, [6] also presented counterexamples where this equivalence breaks down when we move away from GAMs.

The central idea of this work is **to optimize over the integration region** in the FP criterion, rather than fixating on the Euclidean dot product. This leads us to propose the Generalized Franz-Parisi (GFP) criterion (see Definition 2). Our motivation arises from the observation that while the Euclidean distance is natural for GAMs (and spin glass models), it may be inappropriate in other statistical settings (see Section 5). This echoes insights from statistical physics, where non-Euclidean distances are used in models beyond spin glasses [22]. Satisfyingly, this generalization enables a broad equivalence with statistical query (SQ) lower bounds, as shown in Theorem 3.

## A.2 Necessity of assumptions in main theorem

In this section, we comment on the necessity of our assumptions for the GFP-hardness and SQ-hardness equivalence.

### A.2.1 Necessacity of Assumption 1

We first show that there is a strong separation between GFP-hardness and SQ-hardness unless some non trivial lower bound on $\min_{u,v}\langle L_u, L_v \rangle_{\mathbb{Q}}$ is assumed, providing support for our Assumption 1. Indeed, we present here a (very) simple counterexample when one allows for $\min_{u,v}\langle L_u, L_v \rangle_{\mathbb{Q}} = 0$.

Let us define a variant of the following model from [6, Section 4.2.] (assume for simplicity $n$ is a multiple of 10 in what follows):

- Under $\mathbb{P}$, we first sample a $u \sim \text{Unif}(\{x \in \{-1, 1\}^n : \sum x_i = 8n/10\})$. Then under $\mathbb{P}_u$ each sample always equals to $u$, i.e., $\mathbb{P}_u$ is the Dirac measure on $u$.

- Under $\mathbb{Q}$ for each sample, we sample $u \sim \text{Unif}(\{-1, 1\}^n)$ and output $u$.

It is easy to see that for all $u, v \in \{x \in \{-1, 1\}^n : \sum x_i = 8n/10\}$,

$$\langle L_u, L_v \rangle_{\mathbb{Q}} = 2^n \mathbf{1}(u = v).$$

We first prove that the task is *not even* $(1, 1)$-*SQ hard.* Indeed $(1, 1)$-SQ hard implies $\mathbb{E}[|\langle L_u, L_v \rangle_{\mathbb{Q}} - 1|] \leq 1$, but

$$\mathbb{E}[|\langle L_u, L_v \rangle_{\mathbb{Q}} - 1|] \geq |\langle L_u, L_u \rangle_{\mathbb{Q}} - 1|\pi^2(u = v) \tag{14}$$

$$= (2^n - 1)\binom{n}{9n/10}^{-1} = \omega(1). \tag{15}$$

On the contrary, we now show that the task is $(e^{n^{\Theta(1)}}, \infty, O(1))$-*GFP-hard,* more specifically we prove that it is $(m, q, O(1))$-GFP-hard for any sample size $m$ and $q = \binom{n}{9n/10}^{1/2} = e^{n^{\Theta(1)}}$. Indeed, we have $\pi^2(u \neq v) = 1 - q^{-2}$ and therefore to prove $(m, q)$-GFP hardness it suffices

$$\mathbb{E}[\langle L_u, L_v \rangle_{\mathbb{Q}}^m \mathbf{1}(u \neq v)] = O(1).$$

But in fact it even holds $\mathbb{E}[\langle L_u, L_v \rangle_{\mathbb{Q}}^m \mathbf{1}(u \neq v)] = 0$ completing this proof.

### A.2.2 Necessity of a non-trivial information-theory threshold

The second assumption that our equivalence operates under is that $m_{\text{IT}}$ is non-trivial. We now claim that some non-trivial lower bound on $m_{\text{IT}}$ is also necessary for the connection between the notions of GFP-hardness and SQ-hardness, even under Assumption 1.

Indeed, consider the following multisample problem over graphs. Let $n \in \mathbb{N}$, $p = 1 - n^{-1/4}$ and $k = n^{1/3+o(1)}$.

- Under $\mathbb{P}$ we choose a $u$ being a $k$-clique in $K_n$, chosen uniformly at random (we see $u$ as a $k$-vertex set). Then under $\mathbb{P}_u$ one sample consists of the union of a $G(n, p)$ with the $k$-clique on $u$.
- Under $\mathbb{Q}$ one sample is a sample from $G(n, p)$.

We note this is a multi-sample variant of the classic planted clique model [29], but on the (very) dense regime, which has been recently used to establish circuit lower bounds for the model in [**?** ].

Now, even for $m = 1$ the problem is information-theoretically solvable; indeed, under $\mathbb{P}$ there is always a $k$-clique, while under $\mathbb{Q}$ there is no $k$-clique with probability $1 - o(1)$, and a brute force method can distinguish the two cases. Indeed, by a union bound the probability there is a $k$-clique under $\mathbb{Q}$ is at most

$$n^k(1 - n^{-1/4})^{\binom{k}{2}} \leq \exp\left(n^{1/3}\log n - \Theta(n^{2/3-1/4})\right) = \exp\left(-\Theta(n^{5/12})\right) = o(1).$$

Moreover, the model satisfies Assumption 1. One can see this because the model is a PSM and use Lemma 2. Alternatively, one can just directly observe that for any $u$, we have $L_u(G) = \mathbf{1}(u \text{ is a k-clique in } G)p^{-\binom{k}{2}}$. Hence, for any $u, v$

$$\langle L_u, L_v \rangle_{\mathbb{Q}} = p^{-\binom{|u \cap v|}{2}} = (1 - n^{-1/4})^{-\binom{|u \cap v|}{2}} \geq 1.$$

Now, we prove that this PSM is not SQ-hard even for $m = 1$ and $q = 1$, but it is GFP-hard even for $m = n^{\Theta(1)}$-samples.

We first prove that the task *is not* $(1, 1)$-*SQ-hard.* Notice that $(1, 1)$-SQ hardness is equivalent with the condition $\mathbb{E}[|\langle L_u, L_v \rangle_{\mathbb{Q}} - 1|] \leq 1$. But in this context

$$\mathbb{E}[|\langle L_u, L_v \rangle_{\mathbb{Q}} - 1|] \geq |\langle L_u, L_u \rangle_{\mathbb{Q}} - 1|\pi^2(u = v)$$

$$= (1 - n^{-1/4})^{\Theta(k^2)} \binom{n}{k}$$

$$= \exp\left(\Theta(k^2 n^{-1/4}) - \Theta(k \log n)\right)$$

$$= \exp\left(\Theta(n^{5/12})\right) = \omega(1).$$

On the contrary, the task is $(e^{n^{\Theta(1)}}, n^{1/8}, O(1))$-*GFP-hard*. Fix $m$ samples. Notice that for i.i.d. $u, v \sim \pi$, the overlap $|u \cap v|$ follows an $(n, k, k)$-Hypergeometric. Hence by [6, Lemma 6.6.] for any $q > 0$ if $\delta = \log(k^2 q) > 0$ it holds

$$\pi^2(|u \cap v| \geq \delta) \leq k(k^2/n)^\delta \leq k2^{-\delta} = q^{-2}.$$

Hence, to prove GFP-hardness it suffices to prove that $\mathbb{E}[\langle L_u, L_v \rangle_{\mathbb{Q}}^m \mathbf{1}(|u \cap v| \leq \delta)] = O(1)$. Therefore for $q = \exp(n^\alpha)$ for some $\alpha > 0$, and $m = n^{1/8 + o(1)}$,

$$\mathbb{E}[\langle L_u, L_v \rangle_{\mathbb{Q}}^m \mathbf{1}(|u \cap v| \leq \delta)] \leq \mathbb{E}[(1 - n^{-1/4})^{-m\binom{|u \cap v|}{2}} \mathbf{1}(|u \cap v| \leq \delta)]$$

$$\leq (1 - n^{-1/4})^{-m\binom{\delta}{2}}$$

$$= (1 - n^{-1/4})^{-\Theta(m(\log(kq^2))^2}$$

$$\leq \exp\left(\Theta(mn^{-1/4}(\log(kq^2))^2)\right)$$

$$= \exp\left(n^{-1/8 + 2\alpha}\right) = O(1),$$

where the last equality hold say for any $0 < \alpha < 1/16$. Hence, choosing say $q = e^{n^{1/32}}$ concludes the proof.

# B  Equivalence between LD, SQ, and GFP

In this Appendix, we discuss the equivalence between GFP and low-degree (LD) polynomial hardness. This result is obtained by combining the GFP-SQ equivalence from Section 3 with the equivalence between SQ and LD hardness under noise robustness proved by Brennan et al. [8]. We recall and provide a succinct proof of Brennan et al.'s result for completeness.

## B.1  Low-Degree lower bounds definitions

We start by recalling the definition of a low-degree lower bound. The definition is based on the *low-degree likelihood ratio* $L^{\leq D}$, where we recall that $L^{\leq D}$ denotes the projection of the likelihood ratio onto the subspace of degree-at-most-$D$ polynomials.

### B.1.1  Low-Degree Lower Bounds

The following is the standard definition of Low-Degree hardness as originally stated, for example, in [26].

**Definition 6** (Low-Degree Likelihood Ratio). *For $m$ samples, define the squared norm of the degree-$D$ likelihood ratio (also called the "low-degree likelihood ratio") to be the quantity*

$$\mathrm{LD}(D) := \|L_m^{\leq D}\|_{\mathbb{Q}}^2 = \left\| \left( \mathbb{E}_{u \sim \pi} L_u^{\otimes m} \right)^{\leq D} \right\|_{\mathbb{Q}}^2 = \mathbb{E}_{u,v \sim \pi} \left[ \langle (L_u^{\otimes m})^{\leq D}, (L_v^{\otimes m})^{\leq D} \rangle_{\mathbb{Q}} \right] . \quad (16)$$

*For some increasing sequence $D = D_n$, we say that the hypothesis testing problem above is* hard for the degree-$D$ likelihood *or simply $D$-degree hard if* $\mathrm{LD}(D) = O(1)$.

While we direct the reader to [6, Section 1.2] a relation between the Low-degree likelihood ratio and the performance of low-degree algorithms we highlight some key conjectures in the community.

- We expect the class of degree-$D$ polynomials to be as powerful as all $\exp\left(\tilde{\Theta}(D)\right)$-time tests (which is the runtime needed to naively evaluate the polynomial term-by-term). Thus, if $\mathrm{LD}(D) = O(1)$ (or $1 + o(1)$), we take this as evidence that strong (or weak, respectively) detection requires runtime $e^{\tilde{\Omega}(D)}$; see Hypothesis 2.1.5 of [26].

- On a finer scale, we expect the class of degree-$O(\log n)$ polynomials to be at least as powerful as all polynomial-time tests. Thus, if $\mathrm{LD}(D) = O(1)$ (or $1 + o(1)$) for some $D = \omega(\log n)$, we take this as evidence that strong (or weak, respectively) detection cannot be achieved in polynomial time; see Conjecture 2.2.4 of [26].

We emphasize that the above statements are not true in general (see for instance [39] for some discussion of counterexamples) and depend on the choice of $\mathbb{P}$ and $\mathbb{Q}$, yet remarkably often appear to hold up for a broad class of distributions arising in high-dimensional statistics.

### B.1.2  Low Samplewise Degree Lower Bounds

In multisample settings like ours, a similar notion of "samplewise" low degree lower bounds have been considered in [8].

**Definition 7.** *For $d, k \in \mathbb{N} \cup \{\infty\}$ a function $f : (\mathbb{R}^n)^{\otimes m} \to \mathbb{R}$ has samplewise degree $(d, k)$ if it can be written as a linear combination of functions which have degree at most $d$ in each $x_i$ and non-zero degree in at most $k$ of the $x_i$'s (if $d < \infty$ the function is therefore a polynomial).*

Let's state the hardness criterion associated with this samplewise low degree polynomials:

**Definition 8** (Low Degree (LD) Hardness). *We say a "$\mathbb{P}$ versus $\mathbb{Q}$" detection problem is $(m, d, k, \varepsilon)$-LD hard if*

$$\textbf{\textit{LD:}} \qquad \mathbb{E}\left[ \langle (L_u^{\otimes m})^{\leq d,k}, (L_v^{\otimes m})^{\leq d,k} \rangle \right] \leq 1 + \varepsilon. \quad (17)$$

Notice that this notion of $(d,k)$-low degree hardness is the natural generalization to (16). As a point of comparison, $dk$-degree polynomials contain all $(d,k)$-degree polynomials and $(d,d)$-degree polynomials contain all $d$-degree polynomial.

**Remark B.1** (Explaining Remark 3.3). A nice property of the low samplewise-degree degree projection is that it is easy to relate it to $d$-degree projections. Indeed, using a binomial expansion argument (see [8, Claim 3.3.]),

$$\|L_m^{\leq(d,k)}\|_{\mathbb{Q}}^2 = \mathbb{E}_{u,v\sim\pi}\left[\langle(L_u^{\otimes m})^{\leq(d,k)}, (L_v^{\otimes m})^{\leq(d,k)}\rangle_{\mathbb{Q}}\right] = \sum_{t=0}^{m}\binom{m}{t}\mathbb{E}_{u,v\sim\pi}\left[(\langle L_u^{\leq d}, L_v^{\leq d}\rangle_{\mathbb{Q}} - 1)^t.\right]$$

In particular, if $k=1, d=\infty$, since $\mathbb{E}_{u,v\sim\pi}[\langle L_u, L_v\rangle - 1] = \chi^2(\mathbb{P},\mathbb{Q})$ we have

$$\|L_m^{\leq(\infty,1)}\|_{\mathbb{Q}}^2 = 1 + m\chi^2(\mathbb{P},\mathbb{Q}).$$

In particular, notice that the condition $m\chi^2(\mathbb{P},\mathbb{Q}) = O(1)$ discussed in Theorem 2 and Remark 3.3 is equivalent with a samplewise $(\infty,1)$-degree lower bound for the task, i.e., a lower bound against function that are linear combination of functions of one sample at a time. In [8] the authors prove that SQ lower bounds are (almost) equivalent with sample-wise degree lower bounds, therefore it is perhaps no surprise that the condition $m\chi^2(\mathbb{P},\mathbb{Q}) = O(1)$ can be also explained as a (very) weak consequence of any SQ lower bounds against $m$ samples. Indeed, assume a $\mathbb{P}$ versus $\mathbb{Q}$ detection problem is $(q,m)$-SQ hard for any $q$ (even $q=1$). Then setting $A = \text{support}(\pi)^{\otimes 2}$ we have that it must hold $m\mathbb{E}_{u,v\sim\pi}[|\langle L_u, L_v\rangle_{\mathbb{Q}} - 1|] \leq 1$ and therefore

$$m\chi^2(\mathbb{P},\mathbb{Q}) = m\mathbb{E}_{u,v\sim\pi}[\langle L_u, L_v\rangle_{\mathbb{Q}} - 1] \leq 1.$$

## B.2 Unconditional SQ hardness

Before stating the equivalence between the above LD-hardness criterion and SQ-hardness, we define an *Unconditional Statistical Query* (USQ) hardness criterion, which is equivalent to SQ and often appears as a convenient intermediate step in proofs. This hardness measure appeared, often implicitly, in several prior work (e.g., [8]):

**Definition 9** (Unconditional SQ hardness). *We say a "$\mathbb{P}$ versus $\mathbb{Q}$" detection problem is $(m,t)$-unconditional SQ hard for some even $t$ if*

$$\textbf{USQ:} \quad \mathbb{E}\left[\chi_{\mathbb{Q}}(\mathbb{P}_u, \mathbb{P}_v)^t\right] \leq m^{-t}. \tag{18}$$

The USQ criterion removes the conditioning on event $A$ from the SQ criterion, which makes it much easier to manipulate. USQ hardness is essentially equivalent to SQ hardness as stated in the next proposition:

**Proposition 1** (Equivalence USQ and SQ hardness).

  (i) *If a model is $(m,t)$-USQ hard, then it is $(q, m/q^{2/t})$-SQ hard for all integers $q \geq 1$.*

  (ii) *If a model is $(q, m/q^{2/t})$-SQ hard for all integers $q \geq 1$, then it is $(m',t')$-USQ hard for all $t' < t$ and $m' \leq m \cdot 2^{-1/t}(t-t')^{1/t'}$.*

For simplicity, for $t \geq 4$, we can set $t' = t/2$ and $m' = m$ in Proposition 1.(ii). Proposition 1 was proven in [8]. We provide a succinct proof for completeness.

*Proof of Proposition 1.* **USQ hardness implies SQ hardness.** By Hölder's inequality,

$$\mathbb{E}\left[|\langle L_u, L_v\rangle_{\mathbb{Q}} - 1| \,\big|\, A\right] \leq \frac{(\mathbb{E}[|\langle L_u, L_v\rangle_{\mathbb{Q}} - 1|^t])^{1/t} \cdot (\mathbb{E}[\mathbf{1}[(u,v) \in A]])^{1-1/t}}{\pi^2(A)}$$

$$= \left(\frac{\mathbb{E}[|\langle L_u, L_v\rangle_{\mathbb{Q}} - 1|^t]}{\pi^2(A)}\right)^{1/t}.$$

Assuming that we have $(m, t)$-USQ hardness, this implies that for any $q \geq 1$,

$$\sup_{A: \pi^2(A) \geq q^{-2}} \mathbb{E}\left[ |\langle L_u, L_v \rangle_\mathbb{Q} - 1| \, \big| A \right] \leq \frac{q^{2/t}}{m},$$

which establishes the $(q, m/q^{2/t})$-SQ hardness.

**SQ hardness implies USQ hardness.** For convenience, introduce the random variable $X = |\langle L_u, L_v \rangle_\mathbb{Q} - 1|$ with $(u, v) \sim \pi^2$. Assume that we have $(q, m/q^{2/t})$-SQ hardness for all $q \geq 1$. In particular, for all $A$, we have

$$\mathbb{E}[X|A] \leq \frac{1}{\pi^2(A)^{1/t} m}.$$

Using [8, Fact 4.3], we have for every $t > t' > 0$,

$$\mathbb{E}[X^{t'}] \leq \left( 2 \sup_A \pi^2(A) \cdot \mathbb{E}[X|A]^t \right)^{t'/t} \cdot \frac{t}{t - t'} \leq \frac{2^{t'/t}}{m^{t'}} \cdot \frac{t}{t - t'},$$

which establishes $(t', m')$-USQ hardness for any $t' < t$ and $m' = m \cdot 2^{-1/t}(t - t')^{1/t'}$. $\qquad \square$

## B.3 Noise-robust models and SQ-LD equivalence

An advantage of USQ is that it is directly related to Low Degree lower bounds: USQ hardness is equivalent to LD hardness with $d = \infty$, that is, with no degree-constraint on each sample in the projection.

**Proposition 2** (Equivalence between USQ and LD hardness with $d = \infty$).

    *(i) If a model is $(m, \infty, k, \varepsilon)$-LD hard, then it is $(m', k)$-USQ hard with $m' = m/(k\varepsilon^{1/k})$.*

    *(ii) If a model is $(m, k)$-USQ hard, it is $(m, \infty, k, e - 1)$-LD hard. More generally, it will be $(m', \infty, k, em'/m)$-LD hard for all $m' < m$.*

*Proof of Proposition 2.* We follow the proof in [8]. Assume that the model is $(m, \infty, k, \varepsilon)$-LD hard. Then

$$\|\mathbb{E}_u[(L_u^{\otimes m})^{\leq \infty, k}] - 1\|_\mathbb{Q}^2 = \sum_{s=1}^{k} \binom{m}{s} \mathbb{E}_{u,v}[\chi_\mathbb{Q}(\mathbb{P}_u, \mathbb{P}_v)^s] \leq \varepsilon,$$

and in particular, for $k$ even,

$$\|\mathbb{E}_u[(L_u^{\otimes m})^{\leq \infty, k}] - 1\|_\mathbb{Q}^2 - \|\mathbb{E}_u[(L_u^{\otimes m})^{\leq \infty, k-1}] - 1\|_\mathbb{Q}^2 = \binom{m}{k} \mathbb{E}_{u,v}[\chi_\mathbb{Q}(\mathbb{P}_u, \mathbb{P}_v)^k] \leq \varepsilon.$$

This implies that $\mathbb{E}_{u,v}[\chi_\mathbb{Q}(\mathbb{P}_u, \mathbb{P}_v)^k] \leq \varepsilon/\binom{m}{k} \leq \varepsilon k^k/m^k$. On the other hand, $(m, k)$-USQ hardness implies that

$$\|\mathbb{E}_u[(L_u^{\otimes m})^{\leq \infty, k}] - 1\|_\mathbb{Q}^2 \leq \sum_{s=1}^{k} \frac{m^s}{s!} \mathbb{E}_{u,v}[\chi_\mathbb{Q}(L_u, L_v)^s] \leq (e - 1).$$

More generally, we will have for $m' < m$

$$\|\mathbb{E}_u[(L_u^{\otimes m'})^{\leq \infty, k}] - 1\|_\mathbb{Q}^2 \leq \sum_{s=1}^{k} \frac{(m')^s}{s!} \mathbb{E}_{u,v}[\chi_\mathbb{Q}(L_u, L_v)^s] \leq \sum_{s=1}^{k} \frac{1}{s!} \left( \frac{m'}{m} \right)^s \leq e \frac{m'}{m},$$

which concludes the proof. $\qquad \square$

Combining this equivalence of USQ and LD($d = \infty$) with the equivalence between USQ and SQ in Proposition 1, we can directly state an (unconditional) equivalence between SQ and LD($d = \infty$) hardness. In order to transfer this equivalence to LD with $d < \infty$, one can assume that the model with $d = \infty$ and $d < \infty$ are close to each other: this assumption is equivalent to being *noise-robust* (in some sense, see discussions in [8]).

**Assumption 2** (Noise robustness)**.** *We say a "$\mathbb{P}$ versus $\mathbb{Q}$" detection problem is $(d, k, \delta)$-noise robust if*

$$\|\mathbb{E}_u[(L_u^{>d})^{\otimes k}]\|_{L^2(\mathbb{Q})}^2 \leq \delta. \tag{19}$$

Under this assumption, one can state the equivalence between LD and USQ:

**Proposition 3** (Equivalence between LD and USQ Hardness)**.**

(i) *If the model is $(m, t)$-USQ hard, then the model is also $(m', d, k', em'/m)$-LD hard for all $m' \leq m$, $k' \leq t$ and $d \geq 1$.*

(ii) *If the model is $(m, d, k, \varepsilon)$-LD hard and we further assume that it is $(d, k, \delta)$-noise robust, then the model is $(m', k)$-USQ hard with*

$$m' = \frac{m}{m\delta^{1/k} + k\varepsilon^{1/k}}.$$

*Proof of Proposition 3.* Part (i) is directly implied by Proposition 2.(ii). For part (ii), following the same argument as in the proof of Proposition 2.(i), we get

$$\mathbb{E}[|\langle L_u^{\leq d}, L_v^{\leq d} \rangle - 1|^k] \leq \varepsilon \frac{k^k}{m^k}.$$

Then using [8, Lemma 3.4], we obtain

$$\mathbb{E}[|\langle L_u, L_v \rangle - 1|^k]^{1/k} \leq \mathbb{E}[|\langle L_u^{\leq d}, L_v^{\leq d} \rangle - 1|^k]^{1/k} + \mathbb{E}[|\langle L_u^{>d}, L_v^{>d} \rangle|^k]^{1/k}$$

$$\leq \varepsilon^{1/k} \frac{k}{m} + \delta^{1/k} = \frac{k\varepsilon^{1/k} + m\delta^{1/k}}{m},$$

which concludes the proof. $\qquad \square$

Then, the equivalence between LD and SQ hardness in [8] is obtained by combining Proposition 3 and Proposition 1. We state it below for completeness:

**Theorem 5** (Equivalence between LD and SQ hardness)**.**

(i) *If the model is $(q, m/q^{2/t})$-SQ hard for all $q \geq 1$ (with $t \geq 4$, for simplicity), then it is $(m', d, k', em'/m)$-LD hard for all $m' \leq m$, $k' \leq t/2$. and $d \geq 1$.*

(ii) *If the model is $(m, d, k, \varepsilon)$-LD hard and we further assume that it is $(d, k, \delta)$-noise robust, then the model is $(q, m'/q^{2/t})$-SQ hard for all $q \geq 1$ with*

$$m' = \frac{m}{m\delta^{1/k} + k\varepsilon^{1/k}}.$$

### B.4 Equivalence of GFP, SQ, and LD hardness for noise-robust models

Based on the SQ-LD equivalence stated in the previous section (Theorem 5) and the equivalence between GFP and SQ (Theorem 3), we can state an equivalence between GFP and LD hardness for noise-robust models.

**Theorem 6** (LD and $\rho_G$-FP Equivalence)**.** *Suppose a "$\mathbb{P}$ versus $\mathbb{Q}$" task satisfies Assumption 1 for a group $G$.*

(i) *If the model is $(m, d, k, \varepsilon)$-LD hard and we further assume that it is $(d, k, \delta)$-noise robust, then the model is $(q', m', e|G|^{-1}\tilde{m}q^{2/t}/\tilde{m})$-$\rho_G$-FP hard for any integers $q \geq 1$, $q' \leq q/\sqrt{2}$, and $m' \leq \tilde{m}/2$, with*

$$\tilde{m} = \frac{m}{m\delta^{1/k} + k\varepsilon^{1/k}}.$$

*(ii) If a task is $(q, m, \varepsilon)$-$\rho_G$-FP hard for some $q, m$ integers. Assume that there exists an $r = r(q) > 0$ such that $\pi^2(\rho_G(u, v) < r) = 1 - q^{-2}$ and $m$ is even. Then, for all even integer $4 \le t \le \log(q)/\log(m)$, the model is also $(m', d, k', em'/\tilde{m})$-LD hard for all $m' \le \tilde{m}$ and $k' \le t/2$, and $d \ge 1$, where*

$$\tilde{m} = \frac{m}{t(1 + \varepsilon)^{1/t} + \chi^2(\mathbb{P}^{\otimes 4t} \| \mathbb{Q}^{\otimes 4t})}.$$

Note that the implication of GFP hardness to LD hardness is unconditional. LD hardness with $d = \infty$ implies GFP hardness, while for $d < \infty$, this implication holds under the noise robustness assumption.

# C  Proofs of main theorems

## C.1  Proof of Theorem 2

*Proof of Theorem 2.* It is clear that $(q, m, \varepsilon)$-$\rho_G$-FP hard implies $(q, m, \varepsilon)$-GFP$_G$ hard as for the event

$$A := \{\rho_G(u, v) < r(q)\},$$

it clearly holds $\pi^{\otimes 2}(A) \geq 1 - q^{-2}$ and, since $G$ is a group, $G^{\otimes 2}(A) = A$. Hence,

$$\inf_{\substack{A : \pi^{\otimes 2}(A) \geq 1 - q^{-2} \\ G^{\otimes 2}(A) = A}} \mathbb{E}\Big[\langle L_u^{\otimes m}, L_v^{\otimes m}\rangle_{\mathbb{Q}} \mathbf{1}(A)\Big] \leq \mathbb{E}\left[\langle L_u^{\otimes m}, L_v^{\otimes m}\rangle_{\mathbb{Q}} \cdot \mathbf{1}(\rho_G(u, v) < r(q))\right] \leq 1 + \varepsilon,$$

implying the desired result.

We now focus on the other direction. By decomposing the likelihood ratio inner product, we obtain

$$\langle L_u^{\otimes m}, L_v^{\otimes m}\rangle_{\mathbb{Q}} = \left(\langle L_u, L_v\rangle_{\mathbb{Q}} - 1 + 1\right)^m = \sum_{t=0}^{m} \binom{m}{t} \cdot \left(\langle L_u, L_v\rangle_{\mathbb{Q}} - 1\right)^t. \tag{20}$$

Taking expectation over the prior $\pi^{\otimes 2}$ conditioned on *any* event $A$ satisfying $G^{\otimes 2}(A) = A$ and $\pi^2(A) = 1 - q^{-2}$, we have

$$
\begin{aligned}
\mathbb{E}\big[\langle L_u^{\otimes m}, L_v^{\otimes m}\rangle_{\mathbb{Q}} \mid A\big] &= \sum_{t=0}^{m} \binom{m}{t} \cdot \mathbb{E}\big[\left(\langle L_u, L_v\rangle_{\mathbb{Q}} - 1\right)^t \mid A\big] \\
&= \sum_{t=1}^{m} \binom{m}{t} \cdot \left(\mathbb{E}_{g \sim \mathrm{Unif}(G)} \mathbb{E}\big[\left(\langle L_{g(u)}, L_{g(v)}\rangle_{\mathbb{Q}} - 1\right)^t \mid A\big]\right) + 1 \\
&\geq \sum_{t=1}^{\lfloor m/2 \rfloor} \binom{m}{2t} \cdot \left(\mathbb{E}\big[\mathbb{E}_{g \sim \mathrm{Unif}(G)}\left(\langle L_{g(u)}, L_{g(v)}\rangle_{\mathbb{Q}} - 1\right)^{2t} \mid A\big]\right) + 1.
\end{aligned}
$$

where in the second equality, we used that $G$ is a $\pi$-preserving transformation, and for the inequality we use Assumption 1.

Clearly for all $t \geq 0$,

$$\mathbb{E}_{g \sim \mathrm{Unif}(G)}\left(\langle L_{g(u)}, L_{g(v)}\rangle - 1\right)_{\mathbb{Q}}^{2t} \geq |G|^{-1} \rho_G(u, v)^{2t}. \tag{21}$$

Therefore, we further conclude that

$$\mathbb{E}\big[\langle L_u^{\otimes m}, L_v^{\otimes m}\rangle_{\mathbb{Q}} - 1 \mid A\big] \geq |G|^{-1} \sum_{t=1}^{\lfloor m/2 \rfloor} \binom{m}{2t} \cdot \mathbb{E}\big[\rho_G(u, v)^{2t} \mid A\big]. \tag{22}$$

Recall that $r(q)$ satisfies

$$\pi^2((u, v) : \rho_G(u, v) \leq r(q)) = \pi^2(A) = 1 - q^{-2}.$$

Hence, by definition of $r(q)$ we have

$$|G|^{-1} \sum_{t=1}^{\lfloor m/2 \rfloor} \binom{m}{2t} \cdot \mathbb{E}\big[\rho_G(u, v)^{2t} \mid A\big] \geq |G|^{-1} \sum_{t=1}^{\lfloor m/2 \rfloor} \binom{m}{2t} \cdot \mathbb{E}\big[\rho_G(u, v)^{2t} \mid \rho_G(u, v) \leq r(q)\big] \tag{23}$$

$$\geq |G|^{-1} \sum_{t=1}^{\lfloor m/2 \rfloor} \binom{m}{2t} \cdot \mathbb{E}\big[\big|\langle L_u, L_v\rangle_{\mathbb{Q}} - 1\big|^{2t} \mid \rho_G(u, v) \leq r(q)\big]. \tag{24}$$

In addition, we notice that for each even order $2t+1$ with $t = 1, \ldots, \lfloor m/2 \rfloor - 1$, it holds by Lemma 4 that

$$\frac{\binom{m}{2t+1} \cdot \left| \langle L_u, L_v \rangle_{\mathbb{Q}} - 1 \right|^{2t+1}}{\sqrt{\binom{m}{2t} \cdot \left| \langle L_u, L_v \rangle_{\mathbb{Q}} - 1 \right|^{2t} \cdot \binom{m}{2t+2} \cdot \left| \langle L_u, L_v \rangle_{\mathbb{Q}} - 1 \right|^{2t+2}}} = \frac{\binom{m}{2t+1}}{\sqrt{\binom{m}{2t} \cdot \binom{m}{2t+2}}} \leq 2.$$

Therefore, using the inequality $2\sqrt{ab} \leq a + b$ for $a, b \geq 0$, we obtain

$$\binom{m}{2t+1} \cdot \left| \langle L_u, L_v \rangle_{\mathbb{Q}} - 1 \right|^{2t+1} \leq \binom{m}{2t} \cdot \left| \langle L_u, L_v \rangle_{\mathbb{Q}} - 1 \right|^{2t} + \binom{m}{2t+2} \cdot \left| \langle L_u, L_v \rangle_{\mathbb{Q}} - 1 \right|^{2t+2}.$$

Consequently, the right-hand-side of (24) can be further lower bounded by

$$|G|^{-1} \sum_{t=1}^{\lfloor m/2 \rfloor} \binom{m}{2t} \cdot \mathbb{E}\left[ \left| \langle L_u, L_v \rangle_{\mathbb{Q}} - 1 \right|^{2t} \mid \rho_G(u, v) \leq r(q) \right] \tag{25}$$

$$\geq \frac{|G|^{-1}}{3} \sum_{t=2}^{m} \binom{m}{t} \cdot \mathbb{E}\left[ \left| \langle L_u, L_v \rangle_{\mathbb{Q}} - 1 \right|^{t} \mid \rho_G(u, v) \leq r(q) \right] \tag{26}$$

$$\geq \frac{|G|^{-1}}{3} \sum_{t=2}^{m} \binom{m}{t} \cdot \mathbb{E}\left[ \left( \langle L_u, L_v \rangle_{\mathbb{Q}} - 1 \right)^{t} \mid \rho_G(u, v) \leq r(q) \right]. \tag{27}$$

Combining (22), (24), and (27), with the condition of $(q, m, \varepsilon)$-GFP$_\mathcal{T}$ hardness, we obtain

$$\sum_{t=2}^{m} \binom{m}{t} \cdot \mathbb{E}\left[ \left( \langle L_u, L_v \rangle_{\mathbb{Q}} - 1 \right)^{t} \mid \rho_G(u, v) \leq r(q) \right] \leq 3|G| \cdot \mathbb{E}\left[ \langle L_u^{\otimes m}, L_v^{\otimes m} \rangle_{\mathbb{Q}} - 1 \mid A \right]. \tag{28}$$

Again, by the definition of $r(A)$ it follows that

$$\sum_{t=2}^{m} \binom{m}{t} \cdot \mathbb{E}\left[ \left( \langle L_u, L_v \rangle_{\mathbb{Q}} - 1 \right)^{t} \cdot \mathbf{1}(\rho_G(u, v) \leq r(q)) \right] \leq 3|G| \mathbb{E}\left[ \left( \langle L_u^{\otimes m}, L_v^{\otimes m} \rangle_{\mathbb{Q}} - 1 \right) \mathbf{1}(A) \right]$$

and therefore by (20)

$$\mathbb{E}\left[ \left( \langle L_u^{\otimes m}, L_v^{\otimes m} \rangle_{\mathbb{Q}} - 1 \right) \mathbf{1}(\rho_G(u, v) \leq r(q)) \right]$$
$$\leq 3|G| \mathbb{E}\left[ \left( \langle L_u^{\otimes m}, L_v^{\otimes m} \rangle_{\mathbb{Q}} - 1 \right) \mathbf{1}(A) \right] + m \mathbb{E}\left[ \left( \langle L_u, L_v \rangle_{\mathbb{Q}} - 1 \right) \cdot \mathbf{1}(\rho(u, v) \leq r(q)) \right]$$

Next, we aim to upper bound the first order term, namely $m \cdot \mathbb{E}[(\langle L_u, L_v \rangle - 1) \cdot \mathbf{1}(\rho_G(u, v) \leq r(q))]$. Note that $A' := \{(u, v) : \rho_G(u, v) \leq r(q)\}$ is also $G^{\otimes 2}$-invariant. Hence, employing also Assumption 1 we also have

$$\mathbb{E}[(\langle L_u, L_v \rangle - 1) \cdot \mathbf{1}(\rho_G(u, v) \leq r(q))]$$
$$= \mathbb{E}\left[ \left( \mathbb{E}_{g \sim \mathrm{Unif}(G)} \langle L_{g(u)}, L_{g(v)} \rangle_{\mathbb{Q}} - 1 \right) \right) \cdot \mathbf{1}(\rho_G(u, v) \leq r(q)) \right]$$
$$\leq \mathbb{E}\left[ \left( \mathbb{E}_{g \sim \mathrm{Unif}(G)} \langle L_{g(u)}, L_{g(v)} \rangle_{\mathbb{Q}} - 1 \right) \right) \right]$$
$$= \mathbb{E}[(\langle L_u, L_v \rangle - 1)]$$
$$= \chi^2(\mathbb{P}, \mathbb{Q}).$$

Therefore,

$$\mathbb{E}\left[ \left( \langle L_u^{\otimes m}, L_v^{\otimes m} \rangle_{\mathbb{Q}} - 1 \right) \mathbf{1}(\rho_G(u, v) \leq r(q)) \right] \leq 3|G| \mathbb{E}\left[ \left( \langle L_u^{\otimes m}, L_v^{\otimes m} \rangle_{\mathbb{Q}} - 1 \right) \mathbf{1}(A) \right] + m \chi^2(\mathbb{P}, \mathbb{Q}).$$

from which the result follows. $\qquad \square$

**Lemma 4.** *For any $t \in \{1, 2, \ldots, n-1\}$ and $n \geq 3$, we have*

$$\frac{\binom{n}{t}^2}{\binom{n}{t-1} \cdot \binom{n}{t+1}} \leq 4. \tag{29}$$

*Proof.* Note that by the successive ratio between binomial coefficients, we have

$$\frac{\binom{n}{t}^2}{\binom{n}{t-1} \cdot \binom{n}{t+1}} = 1 + \frac{1+n}{t(n-t)} \leq 1 + \frac{1+n}{n-1} = 2 + \frac{2}{n-1} \leq 4. \tag{30}$$

This completes the proof. $\qquad \square$

## C.2 Proof of Theorem 3

*Proof of Theorem 3.* **SQ implies $\rho_G$-FP.** We have that

$$\sup_{A:\pi^2(A)\geq q^{-2}} \mathbb{E}\left[|\langle L_u, L_v\rangle - 1| \,|\, A\right] \leq \frac{1}{m}. \tag{31}$$

Now as $G$ is $\pi$-preserving that easily implies that for any $A$ such that $G^{\otimes 2}(A) = A$, that

$$\sup_{A:\pi^2(A)\geq q^{-2}} \mathbb{E}\left[\rho_G(u,v) \,|\, A\right] \leq |G| \sup_{A:\pi^2(A)\geq q^{-2}} \mathbb{E}\left[\mathbb{E}_{g\sim\mathrm{Unif}(G)} \left|\langle L_{g(u)}, L_{g(v)}\rangle - 1\right| \,|\, A\right] \leq \frac{|G|}{m}.$$

Hence for any $r > 0$ if we set $A_r = \{\rho_G(u,v) \geq r\}$ since $G^{\otimes 2}(A) = A$ we conclude that $\pi^2(A_r) \geq q^{-2}$ implies $r \leq |G|/m$. Recall that $r(q) > 0$ satisfies $\pi^2(A_{r(q)}) \geq q^{-2}$. In particular, $r(q) \leq |G|/m$, and therefore for any $m' \leq m/2$,

$$\mathbb{E}\left[\langle L_u^{\otimes m'}, L_v^{\otimes m'}\rangle_{\mathbb{Q}} \cdot \mathbf{1}(\rho_G(u,v) \leq r(q))\right]$$
$$= \mathbb{E}\left[(\langle L_u, L_v\rangle_{\mathbb{Q}} - 1 + 1)^{m'} \cdot \mathbf{1}(\rho_G(u,v) \leq r(q))\right]$$
$$\leq \mathbb{E}\left[(\rho_G(u,v) + 1)^{m'} \cdot \mathbf{1}(\rho_G(u,v) \leq r(q))\right] \tag{32}$$
$$\leq (r(q) + 1)^{m'} \leq (|G|/m + 1)^{m'} \leq 1 + e|G|m'/m. \tag{33}$$

This concludes the $(q, m', e|G|m'/m)$-$\rho_G$-FP hardness.

**$\rho_G$-FP hardness implies SQ-hardness.** Suppose we have $(q, m, \varepsilon)$-$\rho_G$-FP hardness

$$\mathbb{E}\left[\langle L_u^{\otimes m}, L_v^{\otimes m}\rangle_{\mathbb{Q}} \cdot \mathbf{1}(\rho_G(u,v) < r(q))\right] \leq 1 + \varepsilon, \quad \text{where} \quad \pi^2(\rho_G(u,v) \geq r(q)) = q^2.$$

By definition 4, we have that

$$1 + \varepsilon \geq \mathbb{E}\left[(\langle L_u^{\otimes m}, L_v^{\otimes m}\rangle_{\mathbb{Q}} - 1) \cdot \mathbf{1}(\rho_G(u,v) < r(q))\right]$$
$$= \mathbb{E}\left[\sum_{t=1}^{m} \binom{m}{t} \cdot (\langle L_u, L_v\rangle_{\mathbb{Q}} - 1)^t \cdot \mathbf{1}(\rho_G(u,v) < r(q))\right]$$
$$= \mathbb{E}\left[\sum_{t=1}^{m} \binom{m}{t} \mathbb{E}_{g\sim\mathrm{Unif}(G)}[(\langle L_{g(u)}, L_{g(v)}\rangle_{\mathbb{Q}} - 1)^t] \cdot \mathbf{1}(\rho_G(u,v) < r(q))\right],$$

where the first inequality holds by the definition of the $\rho_G$-FP hardness and the second equality holds by using the elementary $\langle L_u^{\otimes m}, L_v^{\otimes m}\rangle_{\mathbb{Q}} = (\langle L_u, L_v\rangle_{\mathbb{Q}} - 1 + 1)^m$. The last equality holds by using that $G$ is $\pi$-measure preserving. As crucially $\mathbb{E}_{g\sim\mathrm{Unif}(G)}[(\langle L_{g(u)}, L_{g(v)}\rangle_{\mathbb{Q}} - 1)^t] \geq 0$ for all integers $t \geq 0$, we have

$$\mathbb{E}\left[\sum_{t=1}^{m} \binom{m}{t} \mathbb{E}_{g\sim\mathrm{Unif}(G)}[(\langle L_{g(u)}, L_{g(v)}\rangle_{\mathbb{Q}} - 1)^t] \cdot \mathbf{1}(\rho_G(u,v) < r(q))\right]$$
$$\geq \mathbb{E}\left[\sum_{t\leq m,\, t \text{ even}} \binom{m}{t} \cdot \mathbb{E}_{g\sim\mathrm{Unif}(G)}[(\langle L_{g(u)}, L_{g(v)}\rangle_{\mathbb{Q}} - 1)^t] \cdot \mathbf{1}(\rho_G(u,v) < r(q))\right]$$
$$\geq \max_{\substack{1\leq t\leq m,\\ t \text{ even}}} \mathbb{E}\left[\binom{m}{t} \cdot (\langle L_u, L_v\rangle_{\mathbb{Q}} - 1)^t \cdot \mathbf{1}(\rho_G(u,v) < r(q))\right].$$

Hence, combining the two for all even $t$, with $1 \leq t \leq m$,

$$\max_{\substack{1\leq t\leq m,\\ t \text{ even}}} \mathbb{E}\left[(\langle L_u, L_v\rangle_{\mathbb{Q}} - 1)^t \cdot \mathbf{1}(\rho_G(u,v) < r(q))\right] \leq \frac{1 + \varepsilon}{\binom{m}{t}}. \tag{34}$$

Therefore, we have for any even $t$ with $t \leq m$ that

$$\mathbb{E}\big[(\langle L_u, L_v \rangle - 1)^t\big] = \mathbb{E}\big[(\langle L_u, L_v \rangle - 1)^t \mathbf{1}(\rho_G(u, v) < r(q))\big] + \mathbb{E}\big[(\langle L_u, L_v \rangle - 1)^t \mathbf{1}(\rho_G(u, v) \geq r(q))\big]$$

$$\leq \frac{1 + \varepsilon}{\binom{m}{t}} + \mathbb{E}\big[(\langle L_u, L_v \rangle - 1)^{2t}\big]^{1/2} \cdot q^{-1}$$

$$\leq \left( \frac{t(1 + \varepsilon)^{1/t}}{m} + \frac{\chi^2(\mathbb{P}^{\otimes 4t} \| \mathbb{Q}^{\otimes 4t})^{1/2t}}{q^{1/t}} \right)^t.$$

where in the first inequality, we use the Cauchy-Schwarz inequality for the second term and the fact that $\pi^2(\rho_G(u, v) \geq r(q)) \leq q^2$. In the second term, we use the elementary $\binom{m}{t} \geq (m/t)^t$.

Now focusing on $t \leq \log q / \log m$ we further have

$$\mathbb{E}\big[(\langle L_u, L_v \rangle - 1)^t\big] \leq \left( \frac{t(1 + \varepsilon)^{1/t} + \chi^2(\mathbb{P}^{\otimes 4t} \| \mathbb{Q}^{\otimes 4t})}{m} \right)^t. \tag{35}$$

Hence the model is $(\frac{m}{t(1+\varepsilon)^{1/t} + \chi^2(\mathbb{P}^{\otimes 4t} \| \mathbb{Q}^{\otimes 4t})}, t)$-USQ hard. By Proposition 1 we conclude for any $q' > 0$ that the model is $(q', \frac{m(q')^{-2/t}}{t(1+\varepsilon)^{1/t} + \chi^2(\mathbb{P}^{\otimes 4t} \| \mathbb{Q}^{\otimes 4t})})$-SQ hard. The second part follows by setting $t = (\log m)^s$ and $q' = e^{\delta(\log m)^{s+1}}$. $\qquad \square$

# D Details of examples and proofs

## D.1 Symmetric mixed sparse linear regression

The symmetric mixed sparse linear regression (mSLR) setting, is a $\mathbb{P}$ versus $\mathbb{Q}$ detection task defined as follows. Given $k, n \in \mathbb{N}$ with $k \le n$ and $\sigma^2 > 0$, we have:

- Under the planted model, we first sample $u \sim \pi$ uniformly from set $u \in \{0,1\}^n$ with $\|u\|_0 = k$. Then, the sample $(x_i, y_i) \sim \mathbb{P}_u$ is generated by

$$y_i = (k + \sigma)^{-1} \left[ z_i \odot \langle x_i, u \rangle + (1 - z) \odot \langle x_i, -u \rangle + w_i \right],$$

  for independent $w_i \sim \mathcal{N}(0, \sigma^2)$, $x_i \sim \mathcal{N}(0, I_n)$ and $z_i \sim \mathrm{Bern}(1/2)$. Following [5] we also denote $\mathrm{SNR} := k/\sigma^2$.

- Under the null model, the sample $(y_i, x_i) \sim \mathbb{Q}$ is generated by $y_i \sim \mathcal{N}(0, 1)$ and independently $x_i \sim \mathcal{N}(0, I_n)$.

To see that mSLR is a PSM, set for any $u$, $\phi_u := \mathrm{suppport}(u) \cup \{n + 1\}$, that is the coordinates of $((x_i)_j)_{j \in \mathrm{support}(u)}$ and of $y_i$. Then it is easy to confirm that for any subset $S \subseteq \Phi_u$, the marginal distribution $P_u|_{\phi_u}(S)$ does not depend on $u$ but only on $S$; the choice of $\mathrm{suppport}(u) \setminus S$ does not alter this distribution.

It is known that the information theory sample size threshold of the problem is

$$m_{\mathrm{STATS}} = \tilde{\Theta} \left( \frac{k}{\log(\frac{\mathrm{SNR}^2}{2\mathrm{SNR}+1} + 1)} \right),$$

see e.g., [19]. Also in [5] it was proven that in the similar mSLR setting where $u$'s coordinates can take values in $\{-1, 0, 1\}$, if

$$m \le m_{\mathrm{ALG}} = \tilde{\Theta} \left( \frac{(\mathrm{SNR} + 1)^2}{\mathrm{SNR}^2} k^2 \right),$$

then the problem is $O(\log n)$-degree hard. Here we prove that with sample size $m \le (m_{\mathrm{ALG}})^{1-o(1)}$ the problem is also GFP-hard, and hence via Theorem 3 also SQ-hard. Our result holds under a very mild assumption on $\mathrm{SNR}$ being not exponential in $k$. Interestingly, the proof is relatively short.

The first step is to calculate the inner product $\langle L_u^{\otimes m}, L_v^{\otimes m} \rangle_{\mathbb{Q}}$ which accounts to a calculation over the Gaussian measure.

**Lemma 5.** *For any sample size $m$ and any $u, v$ binary $k$-sparse vector, the following holds for the mSLR model:*

$$\langle L_u^{\otimes m}, L_v^{\otimes m} \rangle_{\mathbb{Q}} = \left( 1 - \left( \frac{\langle u, v \rangle}{k + \sigma^2} \right)^2 \right)^{-m} \le \exp \left( \frac{m \langle u, v \rangle^2}{(k + \sigma^2)^2 - \langle u, v \rangle^2} \right).$$

Using Lemma 5 one can prove the GFP-hardness, and therefore the SQ-hardness.

**Theorem 7.** *If $n^{\Omega(1)} = k = o(n^{1/2})$ then for any $m = o\left( \frac{k}{\log(\frac{\mathrm{SNR}^2}{2\mathrm{SNR}+1}+1)} \right)$, it holds*

$$\chi^2(\mathbb{P}^{\otimes m}, \mathbb{Q}^{\otimes m}) = 1 + o(1).$$

*Moreover, for any constant $T > 1$, for any $m = O\left( \frac{(\mathrm{SNR}+1)^2 k^2}{\mathrm{SNR}^2 (\log n)^{2T+2}} \right)$ and $q = e^{\Theta((\log n)^T)}$, the mSLR task is $(q, m, O(1))$-GFP hard. In particular, if $\mathrm{SNR} \le e^{k^{1-\alpha}}$ for some $\alpha > 0$, then for any $T > 1$ the mSLR task is $(e^{\Theta((\log n)^T)}, (\frac{(\mathrm{SNR}+1)^2}{\mathrm{SNR}^2} k^2)^{1-o(1)})$-SQ hard.*

The proof of this Theorem can be found in Appendix D.

### D.1.1 Proofs for mSLR

*Proof of Lemma 5.* Let $\lambda = \sqrt{k/\sigma^2 + 1}$ and since $\langle L_u^{\otimes m}, L_v^{\otimes m}\rangle_{\mathbb{Q}} = (\langle L_u^{\otimes m}, L_v^{\otimes m}\rangle_{\mathbb{Q}})^m$ we focus on the case $m = 1$.

Let $Y = Y_1, X = X_1$. By definition and Bayes' rule,

$$L_u = L_u(X, Y) = \frac{\mathbb{P}(Y|X, u)}{\mathbb{Q}(Y)}$$

Under $\mathbb{Q}$ we have $\lambda\sigma Y \sim \mathcal{N}(0, \lambda^2\sigma^2)$, while under $\mathbb{P}$ conditional on $(X, u)$ we have

$$\lambda\sigma Y = \sqrt{k + \sigma^2}Y \sim \frac{1}{2}\mathcal{N}(\langle X, u\rangle, \sigma^2) + \frac{1}{2}\mathcal{N}(-\langle X, u\rangle, \sigma^2),$$

and so

$$\begin{aligned}
L_u &= \frac{\mathbb{P}(Y|X, u)}{\mathbb{Q}(Y)} \\
&= \frac{1}{2}\lambda\exp\left(-\frac{1}{2\sigma^2}(\lambda\sigma Y - \langle X, u\rangle)^2 + \frac{1}{2\lambda^2\sigma^2}(\lambda\sigma Y)^2\right) \\
&\quad + \frac{1}{2}\lambda\exp\left(-\frac{1}{2\sigma^2}(\lambda\sigma Y + \langle X, u\rangle)^2 + \frac{1}{2\lambda^2\sigma^2}(\lambda\sigma Y)^2\right) \\
&= \frac{\lambda^m}{2}\left\{\exp\left(-\frac{\lambda^2 - 1}{2}Y^2 + \frac{\lambda}{\sigma}Y\langle X, u\rangle - \frac{1}{2\sigma^2}\langle X, u\rangle^2\right)\right. \\
&\quad \left. + \exp\left(-\frac{\lambda^2 - 1}{2}Y^2 - \frac{\lambda}{\sigma}Y\langle X, u\rangle - \frac{1}{2\sigma^2}\langle X, u\rangle^2\right)\right\}.
\end{aligned}$$

Now a standard integration argument using the MGF of the $\chi^2$ distribution (see e.g., the proof of [6, Proposition 6.8.] for an almost identical argument) gives for any $u, v$ binary $k$-sparse vectors,

$$\langle L_u, L_v\rangle_{\mathbb{Q}} = \frac{\lambda^2}{2(2\lambda^2 - 1)^{1/2}}\mathbb{E}_{X\sim\mathbb{Q}}\left(\exp\left(\frac{1}{2\sigma^2(2\lambda^2 - 1)}\left[(1 - \lambda^2)\left(\langle X, u\rangle^2 + \langle X, v\rangle^2\right) + 2\lambda^2\langle X, u\rangle\langle X, v\rangle\right]\right)\right. \tag{36}$$

$$\left. + \exp\left(\frac{1}{2\sigma^2(2\lambda^2 - 1)}\left[(1 - \lambda^2)\left(\langle X, u\rangle^2 + \langle X, v\rangle^2\right) - 2\lambda^2\langle X, u\rangle\langle X, v\rangle\right]\right)\right) \tag{37}$$

Now of course the pair $(\langle X, u\rangle, \langle X, v\rangle)$ follows a bivariate Gaussian law with variances equals to $k$ and covariance $\langle u, v\rangle$. Hence, some standard manipulations (see again the proof of [6, Proposition 6.8.] for an almost identical argument) allow us to derive that for $Z \in \mathbb{R}^{1\times3}$ with i.i.d. $\mathcal{N}(0, 1)$ entries and

$$t := \frac{1}{2\sigma^2(2\lambda^2 - 1)} = \frac{1}{2\sigma^2(2k/\sigma^2 + 1)} = \frac{1}{4k + 2\sigma^2}. \tag{38}$$

it holds

$$\langle L_u, L_v\rangle_{\mathbb{Q}} = \frac{\lambda^{2m}}{2(2\lambda^2 - 1)^{m/2}}\mathbb{E}_Z\left(\exp\left(t\langle M_1, Z^\top Z\rangle\right) + \exp\left(t\langle M_2, Z^\top Z\rangle\right)\right) \tag{39}$$

where for $\ell := \langle u, v\rangle$,

$$M_1 = M_1(\ell) := \begin{pmatrix} 2\ell & \sqrt{\ell(k - \ell)} & \sqrt{\ell(k - \ell)} \\ \sqrt{\ell(k - \ell)} & (1 - \lambda^2)(k - \ell) & \lambda^2(k - \ell) \\ \sqrt{\ell(k - \ell)} & \lambda^2(k - \ell) & (1 - \lambda^2)(k - \ell) \end{pmatrix}.$$

and

$$M_2 = M_2(\ell) := \begin{pmatrix} 2(1 - 2\lambda^2)\ell & (1 - 2\lambda^2)\sqrt{\ell(k - \ell)} & (1 - 2\lambda^2)\sqrt{\ell(k - \ell)} \\ (1 - 2\lambda^2)\sqrt{\ell(k - \ell)} & (1 - \lambda^2)(k - \ell) & -\lambda^2(k - \ell) \\ (1 - 2\lambda^2)\sqrt{\ell(k - \ell)} & -\lambda^2(k - \ell) & (1 - \lambda^2)(k - \ell) \end{pmatrix}.$$

The eigendecompositions of $M_1, M_2$ of the form $\sum_{i=1}^{3} \lambda_i \frac{u_i u_i^\top}{\|u_i\|^2}$ are, first for $M_1$,

$$
\begin{aligned}
u_1^\top &= (0 \quad 1 \quad -1) & \lambda_1 &= (1 - 2\lambda^2)(k - \ell) \\
u_2^\top &= (\sqrt{k-\ell} \quad -\sqrt{\ell} \quad -\sqrt{\ell}) & \lambda_2 &= 0 \\
u_3^\top &= (2\sqrt{\ell} \quad \sqrt{k-\ell} \quad \sqrt{k-\ell}) & \lambda_3 &= k + \ell.
\end{aligned}
\tag{40}
$$

and for $M_2$,

$$
\begin{aligned}
u_1^\top &= (0 \quad 1 \quad -1) & \lambda_1 &= k - \ell \\
u_2^\top &= (\sqrt{k-\ell} \quad -\sqrt{\ell} \quad -\sqrt{\ell}) & \lambda_2 &= 0 \\
u_3^\top &= (2\sqrt{\ell} \quad \sqrt{k-\ell} \quad \sqrt{k-\ell}) & \lambda_3 &= (1 - 2\lambda^2)(k + \ell).
\end{aligned}
\tag{41}
$$

As $t < 1/(4k)$ we have $2t \max\{\|M_1\|_{\mathrm{op}}, \|M_2\|_{\mathrm{op}}\} < (k + \ell)/2k \le 1$. Hence, using [6, Lemma A.5.] for $B(U) = \mathbb{R}^{n \times m}$, we have

$$
\langle L_u, L_v \rangle_{\mathbb{Q}} = \frac{\lambda^2}{2(2\lambda^2 - 1)^{1/2}} (\det(I_3 - 2tM_1)^{-1/2} + \det(I_3 - 2tM_2)^{-1/2}).
\tag{42}
$$

Using (38) and (40), (41) the eigenvalues of the matrices $I_3 - 2tM_1, I_3 - 2tM_2$ are

$$
\{1, 1 - 2t(k + \ell), 1 - 2t(1 - 2\lambda^2)(k - \ell)\} = \left\{1, 1 - \frac{k + \ell}{\sigma^2(2\lambda^2 - 1)}, 1 + \frac{k - \ell}{\sigma^2}\right\}
$$

and

$$
\{1, 1 - 2t(k - \ell), 1 - 2t(1 - 2\lambda^2)(k + \ell)\} = \left\{1, 1 - \frac{k - \ell}{\sigma^2(2\lambda^2 - 1)}, 1 + \frac{k + \ell}{\sigma^2}\right\}.
$$

Since $\lambda^2 = k/\sigma^2 + 1$ we have

$$
\frac{\lambda^2}{\sqrt{2\lambda^2 - 1}} \det(I_3 - 2tM_1)^{-1/2} = \lambda^2 \left[\left(2\lambda^2 - 1 - \frac{k + \ell}{\sigma^2}\right)\left(1 + \frac{k - \ell}{\sigma^2}\right)\right]^{-1/2}
\tag{43}
$$

$$
= \frac{\frac{k}{\sigma^2} + 1}{1 + \frac{k - \ell}{\sigma^2}}
\tag{44}
$$

$$
= \left(1 - \frac{\ell}{k + \sigma^2}\right)^{-1}.
\tag{45}
$$

and by symmetry,

$$
\frac{\lambda^2}{\sqrt{2\lambda^2 - 1}} \det(I_3 - 2tM_2)^{-1/2} = \left(1 + \frac{\ell}{k + \sigma^2}\right)^{-1}.
\tag{46}
$$

Combining the above,

$$
\langle L_u, L_v \rangle_{\mathbb{Q}} = \frac{1}{2}\left(\left(1 - \frac{\ell}{k + \sigma^2}\right)^{-1} + \left(1 + \frac{\ell}{k + \sigma^2}\right)^{-1}\right)
\tag{47}
$$

$$
= \left(1 - \left(\frac{\ell}{k + \sigma^2}\right)^2\right)^{-1}
\tag{48}
$$

$$
\le \exp\left(\frac{m\ell^2}{(k + \sigma^2)^2 - \ell^2}\right),
\tag{49}
$$

where for the last inequality we used that $\log x \ge 1 - 1/x$, for $x > 0$. $\qquad\square$

*Proof of Theorem 7.* We have from the first part of Lemma 5,

$$
\chi^2(\mathbb{P}^{\otimes m}, \mathbb{Q}^{\otimes m}) - 1 = \mathbb{E}_{u,v \sim \pi} \langle L_u^{\otimes m}, L_v^{\otimes m} \rangle_{\mathbb{Q}} \le \mathbb{E}_{u,v \sim \pi} \left(1 - \frac{\langle u, v \rangle^2}{(k + \sigma^2)^2}\right)^{-m}
\tag{50}
$$

But, in this setting $\langle u, v \rangle$ follows an Hypergeometric distribution with parameters $n, k, k$. Hence, by [6, Lemma 6.6],

$$\chi^2(\mathbb{P}^{\otimes m}, \mathbb{Q}^{\otimes m}) - 1 \leq \sum_{\ell=0}^{k} \left(1 - \frac{\ell^2}{(k+\sigma^2)^2}\right)^{-m} \left(\frac{k^2}{n-k}\right)^{\ell} \tag{51}$$

$$\leq \sum_{\ell=0}^{\lfloor k/2 \rfloor} \exp\left(\frac{m\ell^2}{(k+\sigma^2)^2 - \ell^2}\right) e^{-\ell \log(\frac{n-k}{k^2})} + \sum_{\ell=\lfloor k/2 \rfloor}^{k} \left(1 - \frac{k^2}{(k+\sigma^2)}\right)^{-m} e^{-\ell \log(\frac{n-k}{k^2})} \tag{52}$$

$$\leq \sum_{\ell=0}^{\lfloor k/2 \rfloor} e^{\ell m/(k-\ell) - \ell \log(\frac{n-k}{k^2})} + k \left(\frac{(\frac{k}{\sigma^2})^2}{2\frac{k}{\sigma^2} + 1} + 1\right)^m e^{-\Theta(k \log(\frac{n-k}{k^2}))}, \tag{53}$$

where for the last inequality we used $\log x \geq 1 - 1/x$ for $x > 0$.

Since $k^2 = o(n), m = o(\frac{k}{\log(\frac{\text{SNR}^2}{2\text{SNR}+1}+1)})$, SNR $= k/\sigma^2$ we have for large enough $n$,

$$k\left(\frac{(\frac{k}{\sigma^2})^2}{2\frac{k}{\sigma^2} + 1} + 1\right)^m e^{-\Theta(k \log(\frac{n-k}{k^2}))} = e^{-\Theta(k \log(\frac{n-k}{k^2}))} = o(1).$$

Moreover, since $k^2 = o(n), m = o(k)$ we have for large enough $n$,

$$\sum_{\ell=0}^{\lfloor k/2 \rfloor} e^{\ell m/(k-\ell) - \ell \log(\frac{n-k}{k^2})} \leq \sum_{\ell=0}^{\lfloor k/2 \rfloor} e^{2\ell m/k - \ell \log(\frac{n-k}{k^2})} \leq \sum_{\ell=0}^{\lfloor k/2 \rfloor} e^{-\ell \log(\frac{n-k}{k^2})/2} = 1 + o(1).$$

Now, fix any $T > 1$. Notice $\langle L_u, L_v \rangle_{\mathbb{Q}}$ is an increasing function of $\langle u, v \rangle$. Hence, for any $q > 0$ there exists $\delta_0(q) > 0$ such that $\{\rho_{\text{id}}(u, v) \geq r(q)\} = \{\langle u, v \rangle \geq \delta_0(q)\}$. Moreover, from the tail of $\langle u, v \rangle$ which is an $(n, k.k)$ Hypergeometric distribution, there exists some $q = q_T = e^{\Theta((\log n)^T)}$ for which there exists $r(T)$ with $\pi(\{\rho_{\text{id}}(u, v) \geq r(T)\}) = q^{-2}$.

Let us then fix $q = q_T$. Notice that if we choose $\delta := \log(kq^2) = \Theta((\log n)^{T+1})$, then we have for large enough $n$, by [6, Lemma 6.6] $\pi^{\otimes 2}(\langle u, v \rangle \geq \delta) \leq k(\frac{k^2}{n})^{\delta} \leq k2^{-\delta} = 1/q^2$. Hence, $\delta_0 \leq \delta$. Combining the above with the second part of Lemma 5,

$$\mathbb{E}_{u,v \sim \pi}(\langle L_u^{\otimes m}, L_v^{\otimes m} \rangle_{\mathbb{Q}} \mathbf{1}(\langle u, v \rangle \leq \delta_0)) \leq \mathbb{E}_{u,v \sim \pi} \exp\left(\frac{m \langle u, v \rangle^2}{(k+\sigma^2)^2 - \langle u, v \rangle^2}\right) \mathbf{1}(\langle u, v \rangle \leq \delta))$$

$$\leq \exp\left(\frac{m\delta^2}{(k+\sigma^2)^2 - \delta^2}\right)$$

$$= \exp\left(m\Theta(\frac{(\log n)^{2(T+1)}}{k^2(1 + \text{SNR}^{-1})^2})\right) = O(1),$$

for any $m = O(\frac{k^2(1+\text{SNR}^{-1})^2}{(\log n)^{2(T+1)}})$. Hence, the model is $(q = q_T, m = \Theta(\frac{k^2(1+\text{SNR}^{-1})^2}{(\log n)^{2(T+1)}}), O(1))$-$\rho_{\text{id}}$-hard. Using now Theorem 3 for $m_{\text{IT}} = (\log n)^T$, which is permissible to use using our $\chi^2$ bound and that SNR $\leq e^{k^{1-\alpha}}$ for some $\alpha > 0$, we conclude for all $T > 1$, the $(e^{\Theta((\log n)^{T-1})}, (k^2(1 + \text{SNR}^{-1})^2)^{1-o(1)})$-SQ hardness. The result follows. $\qquad\square$

## D.2 Non-Gaussian component analysis

The following model was introduced in [16] to capture the complexity of learning Gaussian mixtures.

**Definition 10** (Non-Gaussian component analysis model). *A "$\mathbb{P}$ versus $\mathbb{Q}$" detection problem is a Non-Gaussian component (NGCA) model if:*

- *There exists $\mu \in \mathcal{P}(\mathbb{R})$ such that, under the planted hypothesis $\mathbb{P}_u$ with $u \in \mathcal{S}^{n-1}$, we sample $x \sim \mathcal{N}(0, I_n)$ and replace the component $\langle x, u \rangle \cdot u$ by $z \cdot u$ where $z \sim \mu$ independently;*

- *Under the null model, we sample $x \sim \mathcal{N}(0, I_n)$.*

In other words, an NGCA model is an isotropic Gaussian distribution with a non-Gaussian marginal in direction $u$. The SQ-hardness for NGCA models established also in [16] has been of big importance in proving many recent SQ-hardness for learning tasks, such as for robust estimation of Gaussians [17] and robust linear regression [17] among others.

Interestingly, we can also connect all NGCA models with GFP-hardness for $G = \mathbb{Z}_2$. By a direct Hermite expansion, we can decompose the likelihood function (in $L^2(\mathbb{Q})$)

$$L_u(x) = 1 + \sum_{i=s^*}^{\infty} \nu_i h_i(\langle u, x \rangle), \qquad \nu_i := \mathbb{E}_{z \sim \mu}[h_i(z)],$$

where $h_i$ is the (normalized) degree-$i$ Hermite polynomial. Here we denoted $s^* > 0$ the first non-zero coefficient $\nu_i \neq 0$, that is, the smallest moment of $\mu$ that disagrees with $N(0,1)$ moments (we call $s^*$ the generative exponent of the NGCA model). The inner-product of the likelihood ratios is given for all $u, v \in \mathcal{S}^{n-1}$ by

$$\langle L_u, L_v \rangle = 1 + \sum_{i=s^*}^{\infty} \nu_i^2 \cdot \left( \langle u, v \rangle \right)^s, \tag{54}$$

where we used that $\mathbb{E}[h_s(\langle u, x \rangle) h_k(\langle v, x \rangle)] = \delta_{ks} \langle u, v \rangle^s$. Similar to GAMs, using again the group $G = \mathbb{Z}_2$ acting on flipping the sign of each parameter, we get

$$\mathbb{E}_{g,g' \sim \text{Unif}(G)}(\langle L_{g(u)}, L_{g'(v)} \rangle_{\mathbb{Q}} - 1) = \sum_{i=s^*, i \text{ even}}^{\infty} \nu_i^2 \cdot \left( \langle u, v \rangle \right)^i \geq 0,$$

concluding that NGCA satisfy Assumption 1 with $G = \mathbb{Z}_2$. Hence, based on our equivalence, for any NGCA model the SQ-hardness is equivalent with the GFP-hardness for any symmetric prior (that is $\pi(-u) = \pi(u)$). We illustrate this equivalence for two standard priors: the uniform prior $\pi = \text{Unif}(\mathcal{S}^{n-1})$ and the $k$-sparse prior $\pi = \text{Unif}(\{u \in \pm\frac{1}{\sqrt{k}}\{0,1\}^n : \|u\|_0 = k\})$.

**Theorem 8** (GFP-hardness of NGCA, uniform prior). *Consider a NGCA model with generative exponent $s^*$ and the uniform prior $\pi = \text{Unif}(\mathcal{S}^{n-1})$. For any $\varepsilon \in (0, 1/2)$, the NGCA model is $(\exp(\Theta(n^\varepsilon)), m, O(1))$-GFP hard with*

$$m = \frac{1}{\nu_{s^*}^2} n^{s^*/2 - \Theta(\varepsilon)}.$$

*Moreover, via our equivalence theorem, the model is $(\exp(n^{\Theta(\varepsilon)}), m^{1-\Theta(\varepsilon)})$-SQ hard.*

**Theorem 9** (GFP-hardness of NGCA, sparse prior). *Consider a NGCA model with generative exponent $s^*$ and the $k$-sparse prior $\pi = \text{Unif}(\{u \in \pm\frac{1}{\sqrt{k}}\{0,1\}^n : \|u\|_0 = k\})$. For any $\varepsilon \in (0, 1/2)$ so that $k = n^{\Omega(\varepsilon)}$, the NGCA model is $(\exp(\Theta(n^\varepsilon)), m, O(1))$-GFP hard with*

$$m = \frac{1}{\nu_{s^*}^2} \min(n^{s^*/2 - \Theta(\varepsilon)}, k^{s^*} n^{-\Theta(\varepsilon)}).$$

*Moreover, via our equivalence theorem, the model is $(\exp(n^{\Theta(\varepsilon)}), m^{1-\Theta(\varepsilon)})$-SQ hard.*

The SQ lower bound in Theorem 8 was proven in [16] by a direct argument: here, we prove this SQ-hardness via equivalence to GFP-hardness. The sparse prior was not considered previously and we include it to illustrate the broad applicability of our equivalence.

### D.2.1 Proofs for Non-Gaussian Component Analysis

*Proof of Theorem 8.* Let us prove that the model is $\rho_{\mathbb{Z}_2}$-FP hard and conclude using the implication in Theorem 2.1. Note that $\rho_{\mathbb{Z}_2}(u, v) = \langle L_u, L_{\text{sign}(\langle u, v \rangle) v} \rangle - 1$, that is

$$\rho_{\mathbb{Z}_2}(u, v) = \sum_{s=s^*}^{\infty} \nu_s^2 |\langle u, v \rangle|^s,$$

and $\rho_{\mathbb{Z}_2}(u, v)$ is an increasing function of $|\langle u, v \rangle|$. Thus, $\rho_{\mathbb{Z}_2}$-FP hardness is equivalent to FP hardness. Using that $\langle u, v \rangle$ under the uniform prior is distributed as the first coordinate of $z \sim \text{Unif}(\mathcal{S}^{d-1})$, we get

$$\pi(|\langle u, v \rangle| \geq \kappa) \leq 2\exp\left(-cn\kappa^2\right),$$

for some universal constant $c > 0$. For simplicity denote $\rho = |\langle u, v \rangle|$. Using that $\nu_s^2 \leq 1$ for any $s \in \mathbb{N}$ by Jensen's inequality, we can write

$$\langle L_u, L_v \rangle - 1 \leq \sum_{s \geq s^*} \nu_s^2 \rho^s \leq \nu_{s^*}^2 \rho^{s^*} + \rho^{s^*+1} \sum_{s \geq s^*+1} \rho^s \leq \rho^{s^*}\left(\nu_{s^*}^2 + \frac{\rho}{1-\rho}\right),$$

so that for $\rho = o_n(1)$ and $n$ large enough, $\langle L_u, L_v \rangle - 1 \leq 2\nu_{s^*}^2 \rho^{s^*}$. We deduce that for $\kappa = n^{-1/2+\varepsilon}$, we have

$$\mathbb{E}\left[\langle L_u^{\otimes m}, L_v^{\otimes m} \rangle_{\mathbb{Q}} \cdot \mathbf{1}(|\langle u, v \rangle| < \kappa)\right] \leq 1 + \sum_{j=1}^m \binom{m}{j} \mathbb{E}\left[(\langle L_u, L_v \rangle_{\mathbb{Q}} - 1)^j \cdot \mathbf{1}(|\langle u, v \rangle| < \kappa)\right]$$

$$\leq 1 + \sum_{j=1}^m (2m\nu_{s^*}^2 \kappa^{s^*})^j.$$

Thus we deduce that the problem is $\left(\exp\left(\Theta(n^\varepsilon)\right), m, \Theta(1)\right)$-GFP hard with $m = n^{s^*/2 - \Theta(\varepsilon)}/\nu_{s^*}^2$.

To use the equivalence with SQ, we need to compute the $\chi^2$-divergence, that is

$$\mathbb{E}[\langle L_u, L_v \rangle^{4t}] = \mathbb{E}[\langle L_u, L_v \rangle^{4t}] = 1 + \sum_{j=1}^{4t} \binom{4t}{j} \mathbb{E}[(\langle L_u, L_v \rangle - 1)^j].$$

Let us bound

$$\mathbb{E}[(\langle L_u, L_v \rangle - 1)^j] = \mathbb{E}[(\langle L_u, L_v \rangle - 1)^j \cdot \mathbf{1}(|\rho| \leq \kappa)] + \mathbb{E}[(\langle L_u, L_v \rangle - 1)^j \cdot \mathbf{1}(|\rho| > \kappa)]$$
$$\leq (2\nu_{s^*}^2 \kappa^{s^*})^j + M^j \exp\left(-cn^{2\varepsilon}\right),$$

where we denoted $M = \|L_u\|_{\mathbb{Q}}^2 - 1 = O(\exp\left(n^{\varepsilon/2}\right))$. Thus

$$\mathbb{E}[(\langle L_u, L_v \rangle - 1)^j] = 1 + \sum_{j=1}^{4t} (8t\nu_{s^*}^2 \kappa^{s^*})^j + (4tM)^j \exp\left(-cn^{2\varepsilon}\right) = O(1),$$

where we used that $t \log(t) = \tilde{\Theta}(n^{\varepsilon/2})$ by assumption. We can therefore apply Theorem 3 with $q' = \exp\left(n^{\varepsilon/2}\right)$ and $t = n^{\varepsilon/2}$ (so that $t \leq \log(q)/\log(m) = \tilde{\Theta}(n^\varepsilon)$). The model is $(q', m')$-SQ hard with

$$m' = \frac{m}{(t(1+\varepsilon)^{1/t} + \chi^2(\mathbb{P}^{\otimes 4t} \| \mathbb{Q}^{\otimes 4t}))(q')^{2/t}} = \Theta(m/t) = m^{1-\Theta(\varepsilon)},$$

which concludes the proof. $\qquad\square$

*Proof of Theorem 9.* The proof proceeds similarly as the proof of Theorem 8. The main difference is the new tail bound on $\langle u, v \rangle$ given in Lemma 6. We now set $\kappa = n^\varepsilon \max(n^{-1/2}, k^{-1})$, so that

$$\pi(|\langle u, v \rangle| \geq \kappa) \leq 2\exp\left(-cn^\varepsilon\right).$$

With this modification, the rest of the proof is identical and we omit it. $\qquad\square$

**Lemma 6** (Tail bound for sparse prior). *Let $u, v$ be independently sampled from the prior $\pi = \text{Unif}(\{u \in \pm\frac{1}{\sqrt{k}}\{0,1\}^n : \|u\|_0 = k\})$. Then for any $t \geq 0$, we have*

$$\pi^2(\langle u, v \rangle \geq t) \leq \exp\left(-c\min\{nt^2, kt\}\right), \tag{55}$$

*for some universal constant $c > 0$.*

## D.3 Single-index Models

Another extremely popular class of models in statistics dating back to the 80s [33, 28] are the so-called single-index models.

**Definition 11** (Single-index model). *A "$\mathbb{P}$ versus $\mathbb{Q}$" detection problem is a Single-index model if:*

- *There exists a distribution $\mu \in \mathcal{P}(\mathbb{R} \times \mathbb{R})$ such that, under the planted hypothesis $\mathbb{P}_u$, we sample $x \sim \mathcal{N}(0, I_n)$ and $y \sim \mu(\cdot|z_u)$, where $z_u := \langle x, u \rangle$;*

- *Under the null model, we sample $x \sim \mathcal{N}(0, I_n)$ and $y \sim \mu_y$, where $\mu_y$ is the marginal distribution of $\mu$.*

Also, all single index models satisfy Assumption 1 for $G = \mathbb{Z}_2$. Indeed, if $s^*$ is the generative exponent of the model [12], following [12] we know that an Hermite expansion gives for some $s^* \in \mathbb{N}$ ($s^*$ is called the generative exponent) that for all $u, v \in \mathcal{S}^{n-1}$,

$$\langle L_u, L_v \rangle_{\mathbb{Q}} = 1 + \sum_{i=s^*}^{\infty} \lambda_i^2 \cdot \left( \langle u, v \rangle \right)^i, \qquad \lambda_i := \|\zeta_i(Y)\|_{\mu_y}, \qquad \zeta_i(y) := \mathbb{E}[h_s(z)|y].$$

From this point on, the argument is identical as in the case of NGCA, including the nonnegativity with $G = \mathbb{Z}_2$ as well as the examples of GFP-hardness with uniform and sparse priors. For completeness, we state separate theorems for single-index models:

**Theorem 10** (GFP-hardness of SI models, uniform prior). *Consider a SI model with generative exponent $s^*$ and the uniform prior $\pi = Unif(\mathcal{S}^{n-1})$. For any $\varepsilon \in (0, 1/2)$, the SI model is $(\exp(\Theta(n^\varepsilon)), m, O(1))$-GFP hard with*

$$m = \frac{1}{\lambda_{s^*}^2} n^{s^*/2 - \Theta(\varepsilon)}.$$

*Moreover, via our equivalence theorem, the model is $(\exp(n^{\Theta(\varepsilon)}), m^{1-\Theta(\varepsilon)})$-SQ hard.*

**Theorem 11** (GFP-hardness of SI models, sparse prior). *Consider a SI model with generative exponent $s^*$ and the $k$-sparse prior $\pi = Unif(\{u \in \pm\frac{1}{\sqrt{k}}\{0,1\}^n : \|u\|_0 = k\})$. For any $\varepsilon \in (0, 1/2)$ so that $k = n^{\Omega(\varepsilon)}$, the SI model is $(\exp(\Theta(n^\varepsilon)), m, O(1))$-GFP hard with*

$$m = \frac{1}{\lambda_{s^*}^2} \min(n^{s^*/2 - \Theta(\varepsilon)}, k^{s^*} n^{-\Theta(\varepsilon)}).$$

*Moreover, via our equivalence theorem, the model is $(\exp(n^{\Theta(\varepsilon)}), m^{1-\Theta(\varepsilon)})$-SQ hard.*

The SQ lower bounds in Theorem 10 and Theorem 11 were proven in [12] and [9] via direct argument. Here, we obtain these bounds via the equivalence of the SQ-hardness and GFP-hardness.

## D.4 Truncated statistics: convex truncation

Learning from truncated data has been a topic of interest since the late 1800s and the pioneering works of Galton [23] and Pearson [35]. Interestingly, there has been some recent line of works on truncated statistics tasks that seeks to revisit these old questions from a computational viewpoint, see e.g., [13], [14] and references therein. In this line of recent work, the problem of detecting a convex truncation in Gaussian noise has been proposed.

**Definition 12.** *Fix $\alpha \in (0, 1)$. A hypothesis testing "$\mathbb{P}$ versus $\mathbb{Q}$" problem is called an $\alpha$-Convex Truncation model if it satisfies:*

1. *Under the null hypothesis $\mathbb{Q}$, $x \sim N(0, I_n)$.*

2. *Under the planted hypothesis $\mathbb{P}_K$, $x \sim N(0, I_n)|K$ where $K$ is a symmetric convex body with Gaussian volume at most $1 - \alpha$.*

Interestingly, also all $\alpha$-Convex Truncation models satisfy Assumption 1 for the trivial group $G$. Perhaps this fact is even more interesting because it turns out Assumption 1 is exactly equivalent with the celebrated Gaussian Correlation Inequality on convex bodies [37, 32].

**Lemma 7.** *Consider an $\alpha$-convex truncated model in Definition 12. For any $K, K'$ two symmetric convex bodies of Gaussian volume $1 - \alpha$, it holds $\langle L_K, L_{K'} \rangle_\mathbb{Q} \geq 1$. This is to say, and $\alpha$-Convex Truncated Model satisfies Assumption 1 for the trivial group.*

*Proof.* For any $K$, it holds $L_K(x) = 1(x \in K)/\mathbb{Q}(K), x \in \mathbb{R}^n$. Hence,

$$\langle L_K, L_{K'} \rangle = \frac{\mathbb{Q}(K \cap K')}{\mathbb{Q}(K)\mathbb{Q}(K')}.$$

But the so-called Gaussian correlation inequality for symmetric convex bodies in convex geometry [37] states exactly that for any symmetric convex bodies $K, K'$ it holds $\mathbb{Q}(K \cap K') \geq \mathbb{Q}(K)\mathbb{Q}(K')$ yielding the result. $\square$

Now, for the $\alpha$-Convex truncation models, the state-of-the-art polynomial-time algorithms require $O(n/\alpha^2)$ samples [15], and the best known information-theoretic lower bound is $\Omega(n/\alpha)$ samples [15]. Using the GFP-hardness to SQ-hardness framework we prove that for some prior on $K$, it is SQ-hard to distinguish with $\tilde{o}(n/\alpha^2)$ samples, providing evidence that the polynomial-time method from [15] cannot be improved.

### D.4.1 A new SQ lower bound

To apply our framework, we focus on the following prior on $K$, a variant of which has been studied in [15] to prove their information-theoretic lower bound of $\Omega(n/\alpha)$ samples. To define it we let

$$K = K_v = \{x \in \mathbb{R}^d : |\langle x, v \rangle| \leq \kappa\},$$

for any $v \in \text{Unif}(\{-1/\sqrt{d}, 1/\sqrt{d}\}^d)$. Here, we choose $\kappa = \kappa(\alpha, d)$ such that the Gaussian measure of each $K_v$ is $1 - \alpha$. Then our prior is uniform among $K_v, v \sim \text{Unif}(\{-1/\sqrt{d}, 1/\sqrt{d}\}^d)$. We refer to the $\alpha$-convex truncation setting with this prior as the *"$\alpha$-Slice Convex Truncation"* model.

We first point out that for any $m = \omega(n/\alpha)$, detection with $m$ samples is always possible in the $\alpha$-Slice Convex Truncation model from a time-inefficient method. Indeed, one can brute-force search for some $v \in \{-1/\sqrt{d}, 1/\sqrt{d}\}^d$ for which it holds: for all $i = 1, 2, \ldots m, |\langle x_i, v \rangle| \leq \kappa$. Under $\mathbb{P}$, there always exists such a vector $v$ and hence the brute force search algorithm will find it with probability 1. Under $\mathbb{Q}$ though a direct union bund gives that such a $v$ exists only with probability at most $2^d(1 - \alpha)^m = o(1)$ for any $m = \omega(d/\alpha)$. Hence, the algorithm can detect with probability $1 - o(1)$. In that context, we prove the following result.

**Theorem 12** ($\rho_{Id}$-FP- and SQ-hardness of Convex Truncation)**.** *Let $n \in \mathbb{N}$ growing and arbitrary $\alpha = \alpha_n \in (0, 1)$. There exists a universal constant $C > 0$ and a prior $\pi$ on the convex bodies $K$ of Gaussian volume $1 - \alpha$ such that for any $q \in \mathbb{N}$ with $q = e^{o(\alpha n)}$, the $\alpha$-Convex Truncation model under $\pi$ is $(q, \frac{Cn}{\alpha^2 \log(1/\alpha)^{3/2} \log q})$-$\rho_{Id}$-FP-hard.*

*In particular, for any constant $T > 0$ if $\alpha = \omega(\frac{(\log n)^T}{n})$ then the $\alpha$-Convex Truncation model under $\pi$ is $(e^{\Theta((\log n)^T)}, \Theta(\frac{n}{\alpha^2 \log(1/\alpha)^{3/2}(\log n)^{2T+1}}))$-SQ hard.*

Satisfyingly the proof of this result is also relatively short. The proof of this Theorem can be found in Appendix D.

### D.4.2 Proofs for Convex Truncation

*Proof of Theorem 12.* Observe that for $L_u := L_{K_u}$ we have via standard Hermite expansion (identical to the argument in [15, Line (32), proof of Claim 24]),

$$\langle L_u, L_v \rangle_\mathbb{Q} = \frac{\mathbb{Q}(K_u \cap K_v)}{(1 - \alpha)^2} = 1 + (1 - \alpha)^{-2} \langle u, v \rangle^2 \left[ \sum_{i=1}^\infty f_{2i}^2 \langle u, v \rangle^{2(i-1)} \right], \tag{56}$$

where $f_i$ is the $i$-th Hermite weight of $\mathbf{1}(x \in [-\kappa, \kappa]), x \in \mathbb{R}$ for $\kappa$ such that $\Phi(\kappa) = 1 - \alpha/2$ where $\Phi$ is the CDF of a standard Gaussian.

Now, conveniently, the authors [27] have already studied the Hermite mass of indicators of symmetric intervals around 0. Indeed, applying [27, Lemma 27] for $j = 2, \theta = k$ imply that

$$f_2^2 = O(\kappa \phi(\kappa)^2),$$

where $\phi$ is the PDF of a standard Gaussian. But observe that by standard tail bounds $\kappa = O(\sqrt{\log(1/\alpha)})$ and from the Mill's ratio bound $\phi(\kappa) = \Theta((1 - \Phi(\kappa))\kappa)$. Combining the above we conclude

$$f_2^2 = O\left(\alpha^2 \log(1/\alpha)^{3/2}\right).$$

Parseval's identity gives $\sum_{i>0} f_i^2 = \alpha(1 - \alpha) \leq \alpha$, and hence for some constant $C > 0$

$$\langle L_u, L_v \rangle_{\mathbb{Q}} \leq 1 + C\left((1 - \alpha)^{-2}\langle u, v\rangle^2 \left(\alpha^2 \log(1/\alpha)^{3/2} + \langle u, v\rangle^2 \alpha\right)\right).$$

Now, notice that from (56), $\langle L_u, L_v \rangle_{\mathbb{Q}}$ is an increasing function of $\langle u, v\rangle^2$. Hence, for any $q > 0$ there exists $\delta_0(q) > 0$ such that $\{\rho_{\mathrm{id}}(u, v) \geq r(q)\} = \{\langle u, v\rangle^2 \geq \delta_0(q)\}$. From Hoeffding's inequality we have that for some constant $C' > 0$ if $\delta = C'\frac{\log q}{n}$ then $\pi^2(\langle u, v\rangle^2 \geq \delta) \leq q^{-2}$. Hence $\delta_0(q) \leq \delta = C'\frac{\log q}{n}$.

Combining the above we have that for any $q = e^{o(\alpha n)}$,

$$\mathbb{E}[\langle L_u, L_v \rangle_{\mathbb{Q}}^m \mathbf{1}(\langle u, v\rangle^2 \leq \delta_0)] \leq \left[1 + C\left((1 - \alpha)^{-2}\delta_0(\alpha^2 \log(1/\alpha)^{3/2} + \delta_0\alpha)\right)\right]^m$$

$$\leq \left[1 + C\left((1 - \alpha)^{-2}C'\frac{\log q}{n}(\alpha^2 \log(1/\alpha)^{3/2} + C'\frac{\log q}{n}\alpha)\right)\right]^m$$

$$\leq \left[1 + 2C\left(C'\frac{\log q}{n(1 - \alpha)^2}(\alpha^2 \log(1/\alpha)^{3/2})\right)\right]^m$$

$$= O(1),$$

as long as $m = O(d/(\alpha^2 \log(1/\alpha)^{3/2} \log q))$. So we conclude the $(q, \Theta(n/(\alpha^2 \log(1/\alpha)^{3/2} \log q)))$-$\rho_{\mathrm{id}}$-FP-hard for any $q = e^{o(\alpha n)}$.

Now via an identical proof to [15, Theorem 23] we have for any $m = o(n/\alpha)$ that

$$\chi^2(\mathbb{P}^{\otimes m}, \mathbb{Q}^{\otimes m}) = O(1).$$

In particular, for any constant $T > 0$,

$$\chi^2(\mathbb{P}^{\otimes (\log n)^T}, \mathbb{Q}^{\otimes (\log n)^T}) = O(1).$$

Finally, notice that again since $\langle L_u, L_v \rangle_{\mathbb{Q}}$ is a strictly increasing function of $\langle u, v\rangle^2$, and $\langle u, v\rangle$ is a sum of iid Rademacher random variables we conclude via standard Central Limit Theorem arguments that for any $T > 0$ there exists $q = q(T) = e^{\Theta((\log n)^{T+1})}$ for which for some $r' = r'(T), r = r(T) > 0$ it holds that $\pi^2(\langle u, v\rangle^2 \geq r) = \pi^2(\rho_{\mathrm{id}}(u, v) \geq r') = q^{-2}$.

Hence, for any $T > 0$ we can apply our equivalence Theorem 3 for $m_{\mathrm{IT}} = (\log n)^T$, $q = q(T + 1)$ (so $\log q = \Theta((\log n)^{T+1})$), and appropriate $t = \Theta((\log n)^T)$ to conclude the $(e^{\Theta((\log n)^T)}, \Theta(\frac{n}{\alpha^2 \log(1/\alpha)^{3/2}(\log n)^{2T+1}}))$-SQ hardness of the task. $\qquad\square$

# E  Details on the GFP-hardness and FP-hardness separation

Below, we provide details on the counterexample described in Section 5.

*Proof of Lemma 3.* By definition, for any $u \in \{0,1\}^{n+1}$,

$$L_u(x) = \prod_{i=0}^{n} \left( \mathbf{1}(x_i = 1) \cdot \frac{1/2 + r \cdot \frac{1-(1-\alpha)\cdot u_i}{2}}{1/2} + \mathbf{1}(x_i = -1) \frac{1/2 - r \cdot \frac{1-(1-\alpha)\cdot u_i}{2}}{1/2} \right)$$

$$= \prod_{i=0}^{n} \left( 1 + rx_i \cdot [1 - (1-\alpha) \cdot u_i] \right).$$

For any $u, v \in \{0,1\}^{n+1}$, the inner product $\langle L_u, L_v \rangle$ satisfies

$$\langle L_u, L_v \rangle = \mathbb{E}_{x \sim \mathbb{Q}} \left[ \prod_{i=0}^{n} \left( 1 + rx_i \cdot [1 - (1-\alpha) \cdot u_i] \right) \left( 1 + rx_i \cdot [1 - (1-\alpha) \cdot v_i] \right) \right]$$

$$= \prod_{i=0}^{n} \mathbb{E}_{x_i \sim \mathrm{Rad}(1/2)} \left( 1 + rx_i \cdot [1 - (1-\alpha) \cdot u_i] \right) \left( 1 + rx_i \cdot [1 - (1-\alpha) \cdot v_i] \right)$$

$$= \prod_{i=0}^{n} \left( 1 + r^2 \cdot (1 - (1-\alpha) \cdot u_i)(1 - (1-\alpha) \cdot v_i) \right)$$

Denote $a_i = 1 + r^2 \cdot (1 - (1-\alpha) \cdot u_i)(1 - (1-\alpha) \cdot v_i)$. When $u_i = v_i = 0$, we have $a_i = 1 + r^2$; when $u_i = v_i = 1$, $a_i = 1 + r^2 \cdot \alpha^2$; when there is exactly one 1 and one 0 in $u_i, v_i$, we get $a_i = 1 + r^2 \cdot \alpha$. We deduce that $a_i = 1 + r^2 \cdot \alpha^{u_i + v_i}$ and the lemma follows. $\square$

Let us consider the $m$-sample version of the hypothesis testing problem. The null hypothesis is then $\mathbb{Q}^{\otimes m}$ and the alternative hypothesis is $\mathbb{E}_{u \sim \pi} \mathbb{P}_u^{\otimes m}$, where $u$ is sampled from the following two-point prior $\pi$:

$$u = \begin{cases} (1, 0, \ldots, 0), & \text{w.p.} \quad \rho, \\ (0, 1, \ldots, 1), & \text{w.p.} \quad 1 - \rho. \end{cases} \tag{57}$$

We abbreviate these vectors as $u_1 = (1, 0, \ldots, 0)$ and $u_2 = (0, 1, \ldots, 1)$ for convenience. Using Lemma 3, it holds that

$$\left\langle L_u^{\otimes m}, L_v^{\otimes m} \right\rangle = \prod_{i=0}^{n} \left( 1 + r^2 \cdot \alpha^{u_i + v_i} \right)^m. \tag{58}$$

Let's next show that this problem is GFP hard but FP easy. Note that $\langle L_u, L_v \rangle \geq 1$ for all $u, v$ and therefore the model verifies Assumption 1 for the trivial group.

**Theorem 13.** *For the two-point prior $\pi$ in (58) with $\rho = \exp\left(-n^\varepsilon/2\right)$, and for $r = n^{-1/2}$, $\alpha = n^{-1+2\varepsilon}$, $m = n^{1-\varepsilon}$ and $D = n^\varepsilon$, where $\varepsilon > 0$ is any small constant, the following hold. The $m$-sample hypothesis testing problem $\mathbb{E}_{u \sim \pi} \mathbb{P}_u^{\otimes m}$ versus $\mathbb{Q}^{\otimes m}$ is $(e^{D/2}, m, \Theta(n^{-\varepsilon}))$-GFP hard but not $(n^{-1}, m, \exp\left(\Theta(n^\varepsilon)\right))$-FP hard. Moreover, via our equivalence theorem the model is $(e^{n^{\Theta(\varepsilon)}}, n^{1-\Theta(\varepsilon)})$-SQ hard.*

*Proof of Theorem 13.* Let us first show it is FP easy. Define $\delta := \delta(n^{-1/2})$ the supremum over $\delta$ such that $\pi^2(\langle u, v \rangle \geq \delta) \geq 1/n$. We observe when $u \neq v$, then we must have $\langle u, v \rangle = 0 < \delta$ by the choice of the two points prior with $\langle u_1, u_2 \rangle = 0$. Therefore,

$$\pi^2(u \neq v) = 2\rho(1-\rho) \leq 2e^{-n^\varepsilon/2} \ll n^{-1} \leq \pi^2(\langle u, v \rangle \geq \delta). \tag{59}$$

We deduce the following lower bound

$$\mathbb{E}_{u,v}[\langle L_u^{\otimes m}, L_v^{\otimes m} \rangle \cdot \mathbf{1}(\langle u, v \rangle < \delta)] \geq \mathbb{E}_{u,v}[\langle L_u^{\otimes m}, L_v^{\otimes m} \rangle \cdot \mathbf{1}(u \neq v)]$$

$$= \pi^2[u \neq v] \cdot \mathbb{E}_{u,v}[\langle L_u^{\otimes m}, L_v^{\otimes m} \rangle \, | \, u \neq v]$$

When conditioned on $u \neq v$, we get

$$\mathbb{E}_{u,v}[\langle L_u^{\otimes m}, L_v^{\otimes m} \rangle \, | \, u \neq v] = (1 + \alpha r^2)^{(n+1)m},$$

by applying Eq. (58), with $u_i + v_i = 1$ for all $0 \leq i \leq n$. Inserting the parameters stated in the lemma, we obtain

$$\mathbb{E}_{u,v}[\langle L_u^{\otimes m}, L_v^{\otimes m} \rangle \cdot \mathbf{1}(\langle u, v \rangle < \delta)] \geq 2\rho(1 - \rho) \cdot (1 + \alpha r^2)^{(n+1)m}$$
$$\geq \exp\left(-\tfrac{1}{2}n^\varepsilon\right) \cdot \left(1 + n^{-2+2\varepsilon}\right)^{(n+1)n^{1-\varepsilon}}$$
$$= \Omega(1) \cdot \exp\left(-\tfrac{1}{2}n^\varepsilon\right) \cdot \exp\left(n^\varepsilon\right) \geq \Omega(1) \cdot \exp\left(\tfrac{1}{2}n^\varepsilon\right).$$

This shows that under our parameter choice, the task is $(n^{-1/2}, m, \exp(\Theta(n^\varepsilon)))$-FP easy.

Let us now show that this model is GFP hard. We will prove that the model is $\rho_{\mathrm{Id}}$-FP hard and conclude using the implication Theorem 2.1. Under the trivial group, we have $\rho_{\mathrm{Id}}(u, v) = \langle L_u, L_v \rangle_{\mathbb{Q}} - 1$. From Eq. (58), the $m$-sample inner product of likelihood ratio is given for $u = v = u_1$ by

$$\langle L_{u_1}^{\otimes m}, L_{u_1}^{\otimes m} \rangle = (1 + \alpha^2 r^2)^m \cdot (1 + r^2)^{mn} \tag{60}$$

and for $u = v = u_2$ by

$$\langle L_{u_2}^{\otimes m}, L_{u_2}^{\otimes m} \rangle = (1 + \alpha^2 r^2)^{nm} \cdot (1 + r^2)^m. \tag{61}$$

Because $\alpha \ll 1$, it is not hard to notice

$$\langle L_{u_1}^{\otimes m}, L_{u_2}^{\otimes m} \rangle < \langle L_{u_2}^{\otimes m}, L_{u_2}^{\otimes m} \rangle < \langle L_{u_1}^{\otimes m}, L_{u_1}^{\otimes m} \rangle. \tag{62}$$

From the definition of $\pi$, it holds that

$$\pi^2(\{u = u_2, v = u_2\} \cup \{u \neq v\}) = 1 - e^{-n^\varepsilon} \geq 1 - q^{-2}, \tag{63}$$

using that we set $q = \exp(D/2)$. Combining with Eq. (62), we conclude that the event $\{\rho(u, v) \leq r(q)\} \subset \{u = u_2, v = u_2\} \cup \{u \neq v\}$. This allows us to estimate the upper bound as

$$\mathbb{E}[\langle L_u^{\otimes m}, L_v^{\otimes m} \rangle \cdot \mathbf{1}(r \leq r(q))] \leq \mathbb{E}[\langle L_u^{\otimes m}, L_v^{\otimes m} \rangle \cdot \mathbf{1}(\{u = u_2, v = u_2\} \cup \{u \neq v\})]$$
$$\leq (1 + \alpha^2 r^2)^{nm} \cdot (1 + r^2)^m. \tag{64}$$

Inserting our choice of parameters, we obtain

$$\mathbb{E}[\langle L_u^{\otimes m}, L_v^{\otimes m} \rangle \cdot \mathbf{1}(r \leq r(q))] \leq \left(1 + n^{-3+4\varepsilon}\right)^{n^{2-\varepsilon}} \cdot \left(1 + n^{-1}\right)^{n^{1-\varepsilon}}$$
$$\leq \exp\left(n^{-1+3\varepsilon} + n^{-\varepsilon}\right) \leq 1 + 2n^{-\varepsilon}. \tag{65}$$

Thus, the model is $(e^{D/2}, m, \Theta(n^{-\varepsilon}))$-$\rho_{\mathrm{Id}}$-FP hard, and therefore $(e^{D/2}, m, \Theta(n^{-\varepsilon}))$-GFP hard.

Finally, let us use the SQ-GFP equivalence in Theorem 3 to show that the model is also SQ hard, with parameters $q' = e^{n^{\varepsilon/2}}$ and $t = n^{\varepsilon/2}$ (where indeed $t \leq \log(q)/\log(m) = \tilde{\Theta}(n^\varepsilon)$). To apply the theorem, we need to compute the $\chi^2$-divergence. Denoting $X = \langle L_u^{\otimes 4t}, L_v^{\otimes 4t} \rangle$ with $t = n^{\varepsilon/2}$,

$$\chi^2(\mathbb{P}^{\otimes 4t}, \mathbb{Q}^{\otimes 4t}) + 1 = \pi^2(u = u_1, v = u_1) \cdot \mathbb{E}[X | u = u_1, v = u_1] + \pi^2(u \neq v) \cdot \mathbb{E}[X | u \neq v]$$
$$+ \pi^2(u = u_2, v = u_2) \cdot \mathbb{E}[X | u = u_2, v = u_2]$$
$$= (1 - \rho)^2 \cdot (1 + \alpha^2 r^2)^{4nt} \cdot (1 + r^2)^{4t} + \rho^2 \cdot (1 + \alpha^2 r^2)^{4t} \cdot (1 + r^2)^{4nt}$$
$$+ 2\rho(1 - \rho) \cdot (1 + \alpha^2 r^2)^{(n+1)4t}$$
$$\leq (1 + n^{-3+4\varepsilon})^{n^{1+\varepsilon}} \cdot (1 + n^{-1})^{n^\varepsilon}$$
$$+ e^{-n^\varepsilon} \cdot (1 + n^{-3+4\varepsilon})^{n^\varepsilon} \cdot (1 + n^{-1})^{n^{1+\varepsilon/2}} + 2n^{-\varepsilon} \cdot (1 + n^{-2+3\varepsilon})^{2n^{1+\varepsilon}}$$
$$\leq 1 + 4n^{-1+\varepsilon}.$$

Thus, we obtain

$$m' = \frac{m}{(t(1+\varepsilon)^{1/t} + \chi^2(\mathbb{P}^{\otimes 4t} \| \mathbb{Q}^{\otimes 4t}))(q')^{2/t}} = m^{1-\Theta(\varepsilon)},$$

and we deduce the model is $(e^{D/2}, m^{1-\Theta(\varepsilon)})$-SQ hard. $\qquad\qquad\square$

