# OpenReview forum: "An Optimized Franz-Parisi Criterion and its Equivalence with SQ Lower Bounds"
_NeurIPS.cc/2025/Conference — NeurIPS 2025 oral_

### Official Review · Reviewer_g7yN · 2025-06-30

**Clarity:** 3
**Significance:** 3
**Originality:** 3
**Rating:** 5
**Confidence:** 4

**Summary:**

This work takes a significant step toward unifying methods for predicting computational hardness in high-dimensional hypothesis testing. The basic setting is as follows. the goal is to distinguish between a family of planted distributions $\mathcal{P} = \\{P_u^{\otimes m} \mid u \in \Theta_n\\}$ and $\mathcal{Q} = Q^\{\otimes m\}$, where m denotes the number of samples and we implicitly consider a sequence of hypothesis testing problems indexed by the data dimension (or instance size) parameter $n$. Note that by assuming a prior $\pi$ over the parameter space $\Theta_n$, the testing effectively becomes a (high-dimensional) simple vs simple testing problem for which likelihood ratio traditionally plays a central role.

There has been a growing recognition that many techniques for predicting computational hardness are closely related, as they rely on common intermediate quantities derived from the likelihood ratios $L_u = P_u/Q$. However, formal connections between these approaches have only recently begun to be established, with the first such unification appearing in the work of Brennan et al. (2021).

This work represents the latest coordinated effort toward unification, this time building connections between the Franz-Parisi potential heuristic from statistical physics and statistical query (SQ) lower bounds from learning theory. In particular, the authors introduce the generalized Franz-Parisi (GFP) criterion and establish its equivalence to a notion of statistical dimension for SQ algorithms, which provides a sufficient (though not necessary) condition for SQ hardness. The criterion is "generalized" because the authors demonstrate that the original FP criterion, introduced in previous work, is overly conservative in its predictions of computational hardness.

The established equivalence also enables a straightforward proof of a new SQ lower bound for the convex truncation problem.

**Questions:**

1. What should one conclude if Assumption 1 does not hold and the GFP and SQ criteria give different predictions of hardness? Which one is a more reliable indicator of computational hardness when equivalence breaks down?
2. The example illustrating the necessity of Assumption 1 in Section A.3.2 appears to be missing some important details. In particular, is there a constraint on the parameters for the planted distributions $P_u$ that ensures the example works as intended?

**Ethical Concerns:**

["NO or VERY MINOR ethics concerns only"]

**Final Justification:**

Rating remains the same. Reasons are provided in my response to the authors' rebuttal.

**Limitations:**

yes

**Quality:**

3

**Strengths And Weaknesses:**

### Strengths

1. **Building bridges between fields that speak different technical languages.** This work situates the GFP criterion within its historical context in statistical physics. By establishing the GFP-SQ equivalence, it connects statistical physics and learning theory, fostering the cross-pollination of intuitions and ideas between these fields.
2. **Simplicity and elegance of the GFP criterion.** The GFP criterion introduced by the authors is a simple, elegant quantity that formalizes a hunch shared by many experts: that existing notions of computational lower bounds, while technically defined differently, largely feel similar in spirit and appear to be different shades of the same "hardness core." By formally establishing the equivalence between GFP and SQ, the work provides a satisfying justification for these intuitions.
3. **Clear and well-organized exposition.** The clear and well-structured exposition makes the main results accessible and allows readers from diverse backgrounds to follow the technical developments.

### Weaknesses

1. **Statistical dimension does not characterize SQ hardness.** The statistical dimension-based criterion is conservative and often predicts problems as "not hard" even when they are, in fact, hard for SQ algorithms. In other words, statistical dimension (Definition 3) provides is a sufficient but not necessary condition for SQ hardness, as noted in [Brennan et al., 2021, Appendix B]:

    > *However, since SDA is **not** a characterization for VSTAT …*
    >

    Hence, referring to Definition 3 as "SQ hardness" can be misleading, especially for readers who may not be aware of this subtlety.

    I suspect that exponentially thin Gaussian pancakes may provide yet another example of the aforementioned gap between statistical dimension and SQ hardness. Notably, Diakonikolas et al. (2024) demonstrated that even pancakes with zero thickness remain SQ-hard.

2. **The role of GFP in the equivalence.** The $\rho_G$-FP criterion appears to be the main workhorse in establishing the equivalence to SQ. Is there reason to believe that GFP is a more fundamental quantity? I agree that the $\rho_G$ criterion seems somewhat ad hoc, as multiple symmetry groups $G$ can correspond to the same high-dimensional testing problem $(\mathcal{P}, \mathcal{Q})$, each giving rise to a different notion of the "overlap potential" $\rho_G$.

**References**

- Ilias Diakonikolas, Daniel Kane, Lisheng Ren, Yuxin Sun. SQ Lower Bounds for Non-Gaussian Component Analysis with Weaker Assumptions. *NeurIPS* 2024.

---

> ### Author Rebuttal · Authors · 2025-07-30
>
> We thank the reviewer for the positive comments on the unification character of our work and our exposition.
>
>
> * **Statistical dimension does not characterize SQ hardness. The statistical dimension-based criterion is conservative and often predicts problems as "not hard" even when they are, in fact, hard for SQ algorithms. In other words, statistical dimension (Definition 3) provides is a su!cient but not necessary condition for SQ hardness, as noted in [Brennan et al., 2021, Appendix B]: However, since SDA is not a characterization for VSTAT … Hence, referring to Definition 3 as "SQ hardness" can be misleading, especially for readers who may not be aware of this subtlety. I suspect that exponentially thin Gaussian pancakes may provide yet another example of the aforementioned gap between statistical dimension and SQ hardness. Notably, Diakonikolas et al. (2024) demonstrated that even pancakes with zero thickness remain SQ-hard.**
>
> This is indeed an important point, and we will make sure to clarify and add comments at multiple places in the text. We will make sure to consistently use the term “SQ-hardness criterion” when referring to the SDA dimension bounds. Of course, the reviewer is correct that an SDA bound is a sufficient condition for SQ-hardness, but not necessary, in the same way the LD-hardness criterion (i.e., LDR bound) criterion is a sufficient but not necessary condition for the failure of low-degree polynomials to achieve strong separation.
>
>
>
> * **The role of GFP in the equivalence. The $\rho$-FP criterion appears to be the main workhorse in establishing the equivalence to SQ. Is there reason to believe that GFP is a more fundamentalquantity? I agree that the criterion seems somewhat ad hoc, as multiple symmetry groups can correspond to the same high-dimensional testing problem , each giving rise to a different notion of the "overlap potential" .**
>
> This is a valuable point. First, we note that the two formulations are equivalent under Theorem 2. We chose to emphasize the optimization-based viewpoint because it more clearly illustrates how our generalized GFP-hardness extends the FP-hardness criterion of Bandeira et al. into a variational framework. This formulation also makes the conceptual advancement more explicit.
>
> Moreover, at least at present, the overlap quantity $\rho_G$ lacks a clear algorithmic interpretation (as we also mention in the Conclusion), and, as the reviewer noted, it may appear somewhat ad hoc. For this reason, we currently prefer to treat $\rho_G$-FP​ as a technical intermediate step rather than the central definition of our framework.
>
> That said, we agree it would be exciting to further explore whether $\rho_G$ carries meaningful algorithmic implications—particularly in relation to local search, landscape geometry, or other structural properties of the underlying models. We see this as a promising direction for future work and that is why we ask it as one of our main questions in the conclusion of the paper.

---

> > ### Comment · Reviewer_g7yN · 2025-08-03
> >
> > Thank you for the clarification and for being receptive to my comment about SQ hardness. Since my questions were only minor clarifications, they did not affect my overall (positive) assessment of the paper. Therefore, my rating remains unchanged.

---

### Official Review · Reviewer_MCz7 · 2025-06-30

**Clarity:** 2
**Significance:** 3
**Originality:** 3
**Rating:** 5
**Confidence:** 4

**Summary:**

This manuscript refines the Franz–Parisi (FP) geometric criterion for average-case computational hardness in high-dimensional inference. The authors introduce a Generalized FP (GFP) criterion that optimizes the choice of “overlap” event and prove an equivalence between GFP-hardness and Statistical-Query (SQ) lower bounds under a mild symmetry assumption . The equivalence is shown to hold for a broad family of detection problems.

**Questions:**

* Title and scope. Will the authors consider adding “Detection” to the title and a short paragraph contrasting annealed FP (detection) with quenched FP (estimation)?

* Clarifying the leap beyond Bandeira et al. Can the authors provide an explicit toy model (in addition to the Section 4 construction) where overlap-based FP passes but GFP detects hardness, and explain the geometric intuition? ALso, in all the exampel provided, can they explain if Bandeira et al applied or fail?. Beyond the Section 4 Rademacher-product example, are there natural models (e.g. permutation-invariant priors) where |〈u,v〉| is irrelevant yet Assumption 1 holds? Including one would strengthen claims of generality.

**Ethical Concerns:**

["NO or VERY MINOR ethics concerns only"]

**Final Justification:**

This paper introduces a refined Franz–Parisi criterion that, by optimizing the overlap event, aligns with SQ hardness and corrects cases where the classical FP criterion fails. The rebuttal clarified the relation to Bandeira et al. (2022), addressed the detection vs. estimation scope, and committed to adding missing references and improving clarity. While additional examples would further enhance intuition, the conceptual advance and broadened applicability are clear. Overall, I find the contribution solid, original, and impactful, and I recommend acceptance.

**Limitations:**

yes

**Quality:**

3

**Strengths And Weaknesses:**

Strengths
- Conceptual advance. Optimizing the overlap event yields a unified hardness criterion that aligns with SQ lower bounds across many canonical detection tasks; this is theoretically neat and practically useful for researchers who favour the SQ framework and unite it with the low degree approach
- Broad theoretical applicability. The mild group-symmetry assumption is verified for several popular models (GAMs, sparse PCA, NGCA, single-index, convex truncation), demonstrating real scope.
- Illustrative counter-example. Section 4 convincingly shows that GFP can succeed where the classical FP criterion misfires, underlining that the refinement is substantive.

Weaknesses

Essentially, the paper is not easy not read, and a lot can be done to enhance clarity:

- Detection vs Estimation not foregrounded. The work targets detection tasks, whereas the original Franz–Parisi potential in physics predicts hardness of estimation. Although this is stated in the abstract, the title and early sections still refer generically to “Franz–Parisi” and "Hardness" which risks confusion. I suggest amending the title (e.g. “An Optimized Franz–Parisi Criterion for Detection”) and maybeadding a short subsection that contrasts annealed FP (used here) with the quenched/estimation version (used in physics and in AMP)?

- Relation to Bandeira et al. (2022) under-explained. Section 4 asserts that GFP and FP diverge, but the presentation is terse and the reader is left unsure why Bandeira’s overlap-based criterion fails in the constructed model. A crisper comparison—perhaps tabulating the two notions side-by-side and walking through a minimal counter-example—would better highlight the novelty. It weould be really a plus in the paper to have more concrete, worked-out examples. The paper claims GFP handles cases where “overlap” is meaningless, yet only one synthetic example is detailed. Additional, simpler examples (even toy 1-D priors) illustrating how the optimized event departs from |〈u,v〉| ≤ ε would help readers grasp the intuition and appreciate originality beyond a minor improvement of Bandeira.

Side note: Additional related-work gap (single-index models).
In discussing the single-index model, the authors cite only Ref. [11] for the characterization of the easy (GE = 1) and hard (GE = 2) phases. Yet these phase diagrams were established several years earlier in two independent lines of work:
• J. Barbier, F. Krzakala, N. Macris, L. Miolane & L. Zdeborová, Optimal errors and phase transitions in high-dimensional generalized linear models, PNAS 116 (2019): 5451 – 5460.
• M. Mondelli & A. Montanari, Fundamental limits of weak recovery with applications to phase retrieval, 2018.

---

> ### Author Rebuttal · Authors · 2025-07-30
>
> We thank the reviewer for their positive comments on the conceptual value and the broad applicability of our work.
>
>
> * **“Essentially, the paper is not easy not read, and a lot can be done to enhance clarity: Detection vs Estimation not foregrounded. The work targets detection tasks, whereas the original Franz–Parisi potential in physics predicts hardness of estimation. Although this is stated in the abstract, the title and early sections still refer generically to “Franz–Parisi” and "Hardness" which risks confusion. I suggest amending the title (e.g. “An Optimized Franz–Parisi Criterion for Detection”) and maybe adding a short subsection that contrasts annealed FP (used here) with the quenched/estimation version (used in physics and in AMP)?”**
>
> We commit to revising the paper to improve as possible its readability and ensure that the scope and significance of our results are clearly stated. (See comment **[1]** in the response to Reviewer Nb5E)
>
> *Detection vs Estimation:* We respectfully disagree that the “estimation versus detection” discrepancy between the classical use of the FP potential for estimation and the detection settings we study is not foregrounded. Very early in the introduction, we comment exactly on this (Lines 40 to 51): *``Given this context, a natural question arises: can one formally connect these two seemingly distinct approaches? At first glance, the answer appears negative, due to a fundamental mismatch in scope. Statistical physics techniques are primarily geared toward estimation problems, where the goal is to recover a hidden signal, while the rigorous frameworks discussed above—such as LD and SQ lower bounds—are focused on detection or hypothesis testing.’’*
> We also highlight that this distinction is a central theme already extensively discussed allso in the work of Bandeira et al. (as we mention on lines 46- 47).
>
>
> *Quenched vs Annealed:* Now, we would also like to note that we discuss in detail the distinction between annealed and quenched version of the FP potential, along with their connection to statistical physics in our Appendix A.2. Yet, this material is not novel as it is already almost identically discussed in Bandeira et al ‘2022 – for this reason it was placed in the appendix primarily because of limited space. That said, we agree that the distinction should be more visible in the main body of the paper and will revise the introduction to include a clearer reference to this appendix—beyond the current brief mention on Lines 61–62—along with a concise summary of the annealed vs. quenched distinction.
>
>
> *Title:* Regarding the title: while the Franz-Parisi potential has traditionally been used in the context of estimation, the “Franz-Parisi criterion” is already formulated only for detection problems in Bandeira et al (2022). For this reason, we do not believe it is necessary to further specify “detection” in the title, which is already quite long. Throughout the paper, we make a careful distinction between the Franz-Parisi criterion and the Franz-Parisi potential, and we define our hardness notion through our generalized FP criterion. We will double-check the manuscript to ensure this distinction is clearly and consistently maintained.
>
>
> * **Relation to Bandeira et al. (2022) under-explained. Section 4 asserts that GFP and FP diverge, but the presentation is terse and the reader is left unsure why Bandeira’s overlap-based criterion fails in the constructed model. A crisper comparison—perhaps tabulating the two notions side-by-side and walking through a minimal counter-example—would better highlight the novelty. It weould be really a plus in the paper to have more concrete, worked-out examples. The paper claims GFP handles cases where “overlap” is meaningless, yet only one synthetic example is detailed. Additional, simpler examples (even toy 1-D priors) illustrating how the optimized event departs from $|\langle u , v \rangle | \leq \varepsilon$ would help readers grasp the intuition and appreciate originality beyond a minor improvement of Bandeira.**
>
>
> The reason our counterexample in Section 4 works to separate the GFP-hardness versus FP-hardness is because the GFP-optimal overlap $\langle L_u,L_v \rangle$ (per Theorem 2) in this model is *not* a function of the overlap $\langle u,v \rangle$. It is this property that causes the FP criterion to fail in this setting, as we explain in the first paragraph of Section 4. We agree that an even simpler separation would be even better, but this was the simplest counterexample we could construct, as most “easy” or lower-dimensional examples (such as 1-d) end up with the overlap $\langle L_u,L_v \rangle$ being a function of $\langle u,v \rangle$.
>
> That being said, we strongly disagree with the reviewer deeming our work a minor improvement of Bandeira et al (2022) from multiple viewpoints. Besides that optimizing the event is an important conceptual advancement to the FP-criterion, which provably corrects it in some cases (such as the model in Section 4), our approach has value when seen also purely as a proof technique as well (independently if FP- and GFP-hardness disagree or not). Specifically it is only under introducing this optimization step that the resulting GFP-hardness can be proven to be equivalent to SQ-hardness for models going well beyond the GAMs that Bandeira et al argued about: including single-index models, NGCA models, planted sparse models and more. This step therefore, greatly expands the scope of this connection between physics-based notions of hardness and TCS notions of hardness.
>
>
> * **Side note: Additional related-work gap (single-index models). In discussing the single-index model, the authors cite only Ref. [11] for the characterization of the easy (GE = 1) and hard (GE = 2) phases. Yet these phase diagrams were established several years earlier in two independent lines of work: • J. Barbier, F. Krzakala, N. Macris, L. Miolane & L. Zdeborová, Optimal errors and phase transitions in high-dimensional generalized linear models, PNAS 116 (2019): 5451 – 5460. • M. Mondelli & A. Montanari, Fundamental limits of weak recovery with applications to phase retrieval, 2018.**
>
>
> We thank the reviewer for providing this reference. We will further add additional references for each model. Note that to the best of our knowledge, reference [11] is the paper that defined the generative exponent for single index models (as the first non-zero coefficient in the Hermite expansion of the likelihood ratio).
>
>
>
> * **Clarifying the leap beyond Bandeira et al. Can the authors provide an explicit toy model (in addition to the Section 4 construction) where overlap-based FP passes but GFP detects hardness, and explain the geometric intuition? ALso, in all the exampel provided, can they explain if Bandeira et al applied or fail?. Beyond the Section 4 Rademacher-product example, are there natural models (e.g. permutation-invariant priors) where $| \langle u,v \rangle |$ is irrelevant yet Assumption 1 holds? Including one would strengthen claims of generality.**
>
>
>
> Bandeira et al showed equivalence between FP-hard and LD-hard for GAMs, and an one-sided implication for Planted Sparse Models where FP-hard implies LD-hard. By equivalence of LD-hard and SQ-hard for noise-robust models, this can be shown to imply equivalence between FP-hard and LD-hard for GAMs, and that FP-hard implies SQ-hard for some noise robust Planted Sparse Models.
>
> In contrast, here, we show equivalence of GFP-hard with SQ-hard for GAMs (by a direct proof),  all Planted Sparse Models (here equivalence instead of one-sided implication), all NGCA, all Single-Index models, and all Gaussian convex truncation models. We believe that our results extend well beyond these examples and that this equivalence can be applied to a wide variety of other problems as well.
>
> About the request of the reviewer for more natural examples compared to Section 4’s counterexample, we do not yet have such an example. Yet, we want to point out that many planted signal models exhibit geometric structures that differ substantially from simple Euclidean overlap $\langle u,v \rangle$, hence, we strongly believe that many such natural examples should exist. For example, in multi-index models, the signal u is an orthogonal matrix. The natural notion of overlap in this case is the Gram matrix $u^T v$, and not for example, their scalar Frobenius inner product. We are actively working on extending our results (and specifically Assumption 1) to apply also for such examples and hope to include additional such applications in the camera-ready version.

---

### Official Review · Reviewer_sUVK · 2025-07-03

**Clarity:** 2
**Significance:** 4
**Originality:** 3
**Rating:** 5
**Confidence:** 3

**Summary:**

The paper explores the computational hardness of statistical inference problems. The paper extends the Franz--Parisi criterion to rigorously characterize hardness beyond the class of Gaussian additive models which was the focus of the existing work. The paper defines a generalized notion of the Franz--Parisi criterion that classifies SQ hardness in a broad class of models beyond the Gaussian additive models.

**Questions:**

1. Is there some intuition why the annealed Franz--Parisi potential is the right object to look instead of the quenched one? It seems possible that in the regimes of parameters than you can show or disprove hardness might coincide with the regimes that quenched and annealed quantities are the same.

2. In what sense is the generalized notion of the GFP hardness optimal? Is it true that if GFP hardness fails, then we also get the absence of SQ hardness? A nice example of the failure of FP criterion to detect SQ hardness is explained in section 4. Is a similar example known if GFP hardness fails to detect hardness in a problem.

3. Is there a simple explanation on how the two notions of hardness in Theorem 2 and Theorem 3 equivalent. Namely the second point makes it a bit hard to parse how the second notion of hardness implies the first due to the extra conditions that appear in the second point.

4. How important is the criterion that $u$ is supported on the sphere on line 55? Is it because if we weakened this condition, then the overlap is not the only quantity we care about, but also the norm of both $u$ and $v$? If that is the case, can we get similar results if we consider the normalized overlap $\frac{\langle u, v \rangle}{\|u\| \|v\|}$ in FP/GFP hardness?

**Ethical Concerns:**

["NO or VERY MINOR ethics concerns only"]

**Final Justification:**

I believe this work is a nice improvement over the original paper that introduced Franz--Parisi hardness criterion. The paper is a bit dense and a lot of the technical results and explanations are deferred to the appendices. I keep my rating as is.

**Limitations:**

Yes

**Paper Formatting Concerns:**

No issues

**Quality:**

4

**Strengths And Weaknesses:**

The main body of this paper provides a nice preview and overview of the original Franz--Parisi criterion and its generalization. However, the nature of the paper is quite technical and much of the proofs are deferred to the appendices. The paper does a adequate job motivating the results and the intuition behind the assumptions, but the constraint on page limit means that the explanations are brief. The appendices are referenced several times in the main body, and perhaps a precise pointer to the relevant section of the appendices to help readers jump to the relevant section.

The connection to detection problems is quite remarkable, as typically from the statistical physics point of view this involves an understanding of the fluctuations. The link with the Franz--Parisi potentials and detection is not obvious at all. The original paper by Bandeira et al is of great importance and this follow-up paper is a strong continuation of this line of work. So although this connection is not new and was established in the earlier work, the extension to statistical query is new and quite interesting as well. The generalization beyond Gaussian additive models is also very powerful.

---

> ### Author Rebuttal · Authors · 2025-07-30
>
> We are grateful to the reviewer for the positive and encouraging feedback on our work.
>
> * **Is there some intuition why the annealed Franz-Parisi potential is the right object to look instead of the quenched one? It seems possible that in the regimes of parameters than you can show or disprove hardness might coincide with the regimes that quenched and annealed quantities are the same.**
>
> ​​We thank the reviewer for this insightful question. Unfortunately, for both the FP criterion and our newly introduced GFP criterion, we are not aware of any alternative explanation for why the annealed FP criterion agrees with the LD/SQ lower bounds, beyond going through the proof itself. We believe that establishing a connection to a criterion based on the *quenched* FP potential is a highly intriguing direction for future research, which we are actively investigating. For this reason, we highlight it as one of the main open questions in the conclusion of our paper.
>
>
> * **In what sense is the generalized notion of the GFP hardness optimal? Is it true that if GFP hardness fails, then we also get the absence of SQ hardness? A nice example of the failure of FP criterion to detect SQ hardness is explained in section 4. Is a similar example known if GFP hardness fails to detect hardness in a problem.**
>
> Our main result shows that, under Assumption 1 and assuming the model of interest has a non-trivial information-theoretic threshold, the SQ and GFP hardness criteria are equivalent. If either of these assumptions is removed, we provide counterexamples in Appendix A3, which (if we understand the question of the reviewer correctly) also provide the requested examples for the reviewer. Specifically, when the problem has a trivial samplewise information-theoretic threshold (e.g., planted clique), the SQ criterion is folklore known to fail to predict hardness correctly, while the GFP criterion gives the correct prediction of hardness (see Appendix A.3.1. for details of such an example). Conversely, if Assumption 1 does not hold, we construct an example where the SQ criterion gives the correct prediction of hardness but the GFP criterion does not (see Appendix A.3.2. for details). Let us know if you need further details on this. We will elaborate more on this in the camera-ready version of our work.
>
>
> * **Is there a simple explanation on how the two notions of hardness in Theorem 2 and Theorem 3 equivalent. Namely the second point makes it a bit hard to parse how the second notion of hardness implies the first due to the extra conditions that appear in the second point.**
>
> We are not entirely certain what the reviewer's question refers to specifically, but we will do our best to further elaborate on Theorems 2 and 3 and their roles. Please let us know if this does not fully address your concerns.
>
> To clarify, Theorems 2 and 3 are intermediate steps in establishing the equivalence between GFP-hardness and SQ-hardness. Specifically:
> 1. *Theorem 2* shows that GFP-hardness is equivalent to a specialized version of GFP-hardness defined with respect to a particular type of event $A$.
> 2. *Theorem 3* then proves that this specialized version is, in turn, equivalent to SQ-hardness.
>
> That said, both theorems rely on a set of technical assumptions that must be satisfied for the total equivalence to hold. We suspect that the number and nature of these assumptions may have caused some confusion, so we summarize and clarify them below, along with why they are reasonable in practice:
> 1. *CDF Condition and Parity of m:* We assume that  $1 - q^{-2}$ lies in the image of the CDF of $\pi^2$, and impose a condition on the parity of $m$. Both are purely technical, and we expect that our results can be generalized to remove them — see Remark 2.2 for further discussion.
> 2. *Bounded Group Size and 1-sample Chi-Squared Divergence:* We assume that  $|G| = O(1)$ and $m \cdot \chi^2(P, Q) = O(1)$. These conditions are satisfied in all natural examples we consider — see again Remark 2.2.
> 3. *Bounded Chi-Squared Divergence with $m_{IT}$ Samples:* We assume that  $\chi^2$ remains $O(1)$ when using $m_{\text{IT}}$ samples. As discussed below Theorem 2, one should think of  $m_{\text{IT}} \sim \log n$, which holds for all examples presented in our paper. Moreover, Appendix A.3.1 shows that this condition is *necessary* for the equivalence, based on a counterexample using the planted clique model.
> 4. *Large Enough $q$:* We assume $q > m^{m_{IT}}$ . In our context,  $q$ corresponds to a “runtime” bound, which is typically taken to be exponential in $n$. Since $m_{IT} \sim \log n$, this condition is typically satisfied in all our examples.
>
>
>
> * **How important is the criterion that  is supported on the sphere on line 55? Is it because if we weakened this condition, then the overlap is not the only quantity we care about, but also the norm of both $u$  and $v$? If that is the case, can we get similar results if we consider the normalized overlap $\langle u,v\rangle /|u| |v|$  in FP/GFP hardness?**
>
> Indeed, we adopted the assumption that the parameter lies on the sphere primarily to simplify notation when comparing our GFP-hardness criterion to FP-hardness; it is not required for the GFP-to-SQ hardness equivalence. However, for FP-hardness, the assumption does play a role. In fact, Bandeira et al. (2022) assume that the parameter has a bounded norm in their proof of the FP-to-LD hardness equivalence, where this assumption is essential for their proof of equivalence to work. We will clarify this point and add a corresponding remark in the revised version of the paper.

---

> > ### Comment · Reviewer_sUVK · 2025-08-04
> >
> > Thank you for the detailed explanations. I look forward to future works on the quenched FP potential. I maintain my original positive assessment of this paper.

---

### Official Review · Reviewer_NyN2 · 2025-07-03

**Clarity:** 3
**Significance:** 3
**Originality:** 4
**Rating:** 5
**Confidence:** 3

**Summary:**

The paper studies information-computation tradeoffs in the context of SQ lower bounds and the Franz-Parisi (FP) criterion. According to the FP criterion (inspired by work from statistical physics) a hypothesis testing problem of distinguishing between a null hypothesis and another distribution family ${P_v}$ parameterized by a (random according to some prior) vector is called FP-hard (with parameters $m,q,\epsilon$) if the average inner product of m-sample likelihood ratios between two members $P_u,P_v$ of the alternative distribution family, where the average is over the a "typical" region for the inner product of $u$ and $v$ is upper bounded by $1+\epsilon$. Prior work (e.g., Bandeira et al. 2022) showed that FP-hardness captures SQ-hardness (in both directions) for Gaussian additive models, and partially for sparse planted models, but also gave counterexamples where the equivalence fails. The present paper proposes a Generalized Franz-Parisi (GFP) criterion. The modification, provided in Definition 2 consists of relaxing the notion of "typical" event to use a more general condition instead of the standard inner product. The paper shows that this is now is equivalent to bound on the Statistical Query dimension. The result relies on an assumption (Assumption 1) which is a correlation condition  required to hold over a group that preserves the prior distribution.

The main equivalence theorem comes in two steps where the authors first show that the "typical" event in the GFP criterion takes a specific form, and then show that the resulting simplified form of the GFP criterion is equivalent to bound on the SQ dimension bound. The combination yields the main result in Theorem 3. The final result also requires the hypothesis problem to be have information-theoretic optimal sample complexity that is lower bounded. A convenient simplified version of the theorem is that if the problem requires $m = \omega(\log n)$ samples information-theoretically, then super-polynomial SQ lower bounds are equivalent to GFP hardness with $q$ (the proxy for runtime) being super-polynomial.

Finally, the paper demonstrates that several well-studied statistical models satisfy the correlation assumption, hence the equivalence applies. These include:

- Gaussian additive models
- Planted sparse models (which includes Gaussian sparse regression, sparse phase retrieval  and sparse PCA) for which the authors show new SQ and GFP-hardness results.
- Non-Gaussian component analysis to which many statistical problems like GMMs estimation and robust mean/regression are reducible to
- Single index models
- Gaussian convex truncation models, for which the paper shows a connection to the correlation inequality for convex bodies in probability theory

**Questions:**

Is there any intuition for understanding the role of $k$ in Assumption 1?

**Ethical Concerns:**

["NO or VERY MINOR ethics concerns only"]

**Final Justification:**

The authors have provided clear explanations to my points. After reviewing the other rebuttals and discussions my view of the paper remains positive, I thus keep my original score.

**Limitations:**

yes

**Paper Formatting Concerns:**

No issues

**Quality:**

3

**Strengths And Weaknesses:**

Strengths:

- There’s significant ongoing interest in characterizing computational-statistical tradeoffs, and this paper builds meaningfully on that direction.
- The generality of the main result is a positive aspect of the paper. Also the paper includes several applications and examples/counter examples to demonstrate why assumptions are needed.

Weaknesses:

- All of the technical work is deferred to the appendix, which makes the main body feel light and the appendix a bit dense.

Overall, I do not have any major concerns about the paper and I would recommend acceptance.

---

> ### Author Rebuttal · Authors · 2025-07-30
>
> We thank the reviewer for their time in reviewing our work and overall positive feedback.
>
>
> * **All of the technical work is deferred to the appendix, which makes the main body feel light and the appendix a bit dense.**
>
> We will reorganize the paper a bit and make an effort to include more details in the main text. See comment **[1]** in response to Reviewer Nb5E.
>
>
> * **Is there any intuition for understanding the role of $k$ in Assumption 1?**
>
> The reason is highly technical. In short, the proof of the equivalence proceeds by an appropriate Taylor expansion of functions of $\langle L_u,L_v \rangle$ and hence boils down to controlling all moments of $\langle L_u,L_v \rangle$. Assumption 1 allows us to work around a critical such bound. Note that in our applications, $G$ is either the trivial group or $Z_2$ and it is sufficient to check this inequality for $k=1$, as explained in Remark 2.1.

---

> > ### Comment · Reviewer_NyN2 · 2025-08-05
> >
> > Thank you for your response. I do not have further questions.

---

### Official Review · Reviewer_Nb5E · 2025-07-17

**Clarity:** 2
**Significance:** 3
**Originality:** 3
**Rating:** 3
**Confidence:** 4

**Summary:**

The authors propose a refinement of the Franz Parisi criterion for the detection of planted signals and the understanding of statistical to computational thresholds introduced by Bandeira et al. in 2022. They show that this optimized criterion is equivalent to Statistical Query lower bounds (which according to the authors can be considered as a representative approach which closely aligns with best known algorithms, together with the low degree polynomial lower bounds)

**Questions:**

Section 1

- lines 51 - 62, you should give an example of a detection problem (even the simple rank one detection problem) illustrating the meaning of P, Q u and v.
- lines 52-56 are not very clear. Why not give an example. Isn’t the planted distrbution the same as the null distribution except that the parameters are updated. E.g. In the additive model with Gaussian noise, if I’m not wrong, the addition of a planted signal corresponds to changing the mean and variance of the original distribution doesn’t it?
- line 58: what are vanishing type I or type II errors ?
- line 63 - 65, in definition 1, from what I understand, the expectation is taken with respect to u and v? I would explicitly indicate this.
- Definition 1 again. From what I understand, you only look at levels of overlap that can be demonstrated for a sufficiently large number of solutions. But then you average over solutions that are more different than the level of overlap delta
- Lines 65-67, The likelihood ratio (dP_u/dQ) should be properly introduced. I strongly encourage a complete introduction such as in e.g. Kunisky et al. “Notes on Computational Hardness of Hypothesis Testing: Predictions using the Low-Degree Likelihood Ratio” (Definition 1.7). I also encourage mentioning the annealed upper bound (and its connection to the FP potential) in the main paper.
- Again on the likelihood ratio, I really recommend moving (11) from appendix A.2 to section 1 and properly introduce the notation L_u^{\times m} in relation to the independence of the Y_i
- The definition of the Franz Paris potential that you give in appendix A2, line 945, is also quite vague.  What does the Potential represent? There is no intuition given. Moreover several explanations are missing. E.g. why do you need the Franz Parisi criterion to be less than 1+epsilon. You should explain somewhere that an O(1) FP criterion corresponds to a regime in which string detection is impossible and that a 1+o(1) FP criterion corresponds to a regime in which weak detection is impossible. Perhaps also indicate that FP is always larger than 1. Perhaps just indicate that the smaller the FP criterion, the harder the problem. I.e. Honestly I think you should at least add part of Bandeira et al 2022, section 1.1. (the reminders on the likelihood ratio and the importance of the squared norm of this ratio)
- it seems that in Equation (1) the inner product should be <L_u, L_v >_Q
- line 68: I really think that appendix A.2. should be moved to section 1 or that you should at least give a better introduction to the notations in Definitions 1 and 2. The notations L_u, L_v really come out of the blue which makes the understanding quite difficult.
- lines 70-73, if you want to provide an additional explanation for (1)-(2) it is good. But your explanation should bring more information. What you do here is you merely describe the formulation. In fact you could do more simple: The left-hand side … corresponding to the constraint. —> this is just a description of what’s already given in (1)-(2). Now if you really want to add a sentence make it simpler: “”we integrate over the region <u,v><=\delta.
- To me, lines 68 to 77 do not bring a lot more information. Except for the sentence on the landscape.
- line 89: “and an one-sided” —> “and a one-sided”?

Section 1.1.

- line 95: “Our main contribution of this work” —> “The main contribution of this work” or “Our main contribution” ?
- lines 98 - 100, you use arguably twice, perhaps replace of the occurrences by another word?
- In (3) it is not clear what \pi represents. \pi seems to be a distribution over u and v? so that \pi^{\otimes 2} is the distribution over independent (u,v). This should be clarified
- In Definition 3, again, no intuition is given.
- Informal statement of Theorem 1: “which we assume (1) it satisfies … and (2) it is…” —> “which we assume (1) satisfies.. and (2) is” ?
- line 135 : “On top that ” —> “On top of that”
- lines 142 - 161, you list examples of tasks that satisfy your assumption before introducing those models. You mention the models in section 1.1 but the reader has to wait for section 3 for those models to be clearly defined. I strongly recommend moving lines 142 to 166 to section 3.

Section 2

- Equation (5), what is the meaning of Unif(G) ?
- line 187: “over the a group” —> “over a group orbit that…”?
- line 189: Isn’t the expecation already applied through the 1/4 and the average over the group? I.e shouldn’t the equation read as 1/4(<L_u, L_v> +<L_{-u}, L_v> + ….) ?
- line 191 : “This averaging approach allows for much greater generality” —> why ? this is not clear to me. Expand or remove.
- line 193 - 194 : “in our examples Section 3” —> “In our examples in section 3”?
- Generally speaking, I would recommend moving Remark 2.1. to section 3 as it does not improves readability of section 2.
- lines 203 - 204 : “Given a group G … it turns out that ..” —> why ? again this it not rigorous.
- lines 217 - 219: “First both the requirement of the parity…” this should go. Either you do it or don’t. But if you don’t, mentioning it reduces readability and should be considered as noise.
- The dependence on the cardinality of the group G is quite interesting and to my opinion would deserve some additional comments.

Section 3

- lines 257 - 258, I would recall 189 - 190 to improve readability ?
- line 272 : “and obtain internet new hardness results” —> Here I’m not sure what you mean
- lines 280 - 283: your definition of the non Gaussian Analysis model is not clear/self contained. In particular what is meant by “replace the component <x, u> \cdot u” by z\cdot u. Is z\cdo u an inner product ? then why not write <z, u> and if it is not, then why not write zu ? also where should one replace the component <x, u> u by zu? The model is not clearly defined. In fact, here again I would advocate less details to gain clarity on the reduced information you provide. By wanting to show more, you lose the readers on everything you want to explain. Is it really necessary to detail all the models in section 3? I would focus on one or two (and perhaps briefly indicate that it works in the other settings as well, without entering into the details)

Bibliography and Appendices

- It is a detail but it would be convenient to add hyperlinks to the bibliographic references.
- The references of the supplementary material do not match the references in the abbreviated paper.

Appendix A.2.

- line 941. Why do you add variable v in the definition of the posterior, why not keeing u as the only variable for the planted signal?
- line 947 : “of the loca dynamics” —> “of the local dynamics”?
- line 971 : “one get” —> “one getS”
- line 977: “as we elaborated in the Introduction ” —> “as we elaborated in the introduction”
- line 980 “it is mathematical connection” —> “it is A mathematical connection”?
- In appendix A.2. lines 937 to 973, if feel there is some confusion with the distributions. In 945 - 946 u has distribution p and v has distribution \nu. In 957 -958, u has distribution p and v has distribution \pi. Finally in 971-972 both u and v have distribution p.

**Ethical Concerns:**

["NO or VERY MINOR ethics concerns only"]

**Limitations:**

see above

**Quality:**

3

**Strengths And Weaknesses:**

- The paper engages with a subject that is particularly relevant in the current context but at this point, it lacks sufficient clarity and organization. I believe it would be more reasonable to take some time to carefully organize it and consider submitting to a journal which favors longer expositions. The main focus of the paper is the Franz Parisi criterion but this criterion is not properly introduced. The idea is great and the paper demonstrates a wide scientific culture but in order for the full depth and implications of this idea to be thoroughly understood, the foundations have to be more clear. In short, the idea is very interesting but it must be reworked (with the objective for the paper to be self contained which is currently not the case). New results are often messy and such messiness should not be a criterion for how novel and influential an idea is. That being said, communication also matters and at this point my feeling is that the structure of the paper does not facilitate a clear understanding of the idea.
Most of the text at this point barely consists of descriptions of the formulas. Central to the generalized Franz Parisi criterion is the notion of group. But the intuition behind this novelty is never fully discussed. Another important notion which is not clearly introduced is the product <L_u, L_v>. Most of the definitions involve tuples of parameters (e.g. the Generalized Franz Paris criterion is defined with respect to a triple (q, m, epsilon)) I think it would be good to recall the meaning of those parameters from time to time in the text. E.g. In lines 107 - 109, the authors give an intuition for q.

- I find it surprising that the Generalized Franz Parisi criterion is group dependent. I feel that this should be further addressed. For example in section 3 you indicate that GAM are FP hard with respect to the group Z_2. Then you say that planted sparse models satisfy assumption 1 w.r.t. the trivial group.. Perhaps I’m missing something but shouldn’t there be a relation. I.e. if a problem is hard with respect to a group, shouldn’t it be hard with respect to the other groups as well ? and if it is not the case, why?


- I think the paper should be reorganized around a practical example such as the Gaussian additive model. At each step, one could then illustrate the different concepts based on this model. It is a bit disappointing given how important the result is and given that the main assumption of the Theorem is satisfied for those models, to not have a clear illustration with more intuition.

- The theoretical definitions 2 and 3 are interesting but better intuition should be provided

---

> ### Author Rebuttal · Authors · 2025-07-30
>
> We thank the reviewer for their positive comments on the novelty of our ideas and the interdisciplinary nature of the work, and for the detailed feedback. We commit to correcting all found minor and typographic mistakes.
>
> **[1]** Several reviewers raised questions about clarity and organization. *In the revision, we will make a serious effort to reorganize the main text and the appendix, with the goal to include more details in the main text.* In particular, (1) we will move more details on the examples from the appendix to the main text, (2) we will ensure that all key assumptions and concepts are properly introduced and in a timely manner in the main body of the paper, and (3) we will also more clearly highlight our contributions compared to Bandeira et al.
>
>
> Below are our response to reviewer’s comments and questions:
>
>
> * **``....take some time to carefully organize it and consider submitting to a journal which favors longer expositions.''**
>
> We respectfully disagree with the reviewer’s suggestion that our work is better suited for a journal. The primary aim of our paper is to introduce a novel criterion of hardness, which is inherently speculative and exploratory in nature, to uncover a perhaps surprising equivalence of this criterion with SQ lower bounds, and to explore the implications of this connection. Our intention is to stimulate discussions within the community and to highlight the deep interconnections among different schools of thought on computational hardness in statistics and machine learning. We believe that such conceptual and thought-provoking contributions are more appropriately presented at a conference, which traditionally encourages innovative and forward-looking work, rather than in the often more formal and conclusive setting of a journal.
>
>
> * **“The main focus of the paper is the Franz Parisi criterion but this criterion is not properly introduced.”**
>
> The Franz-Parisi criterion is introduced in Definition 1 in page 2 of our submission. We are not sure what the reviewer means with this comment.
>
>
> * **“with the objective for the paper to be self contained which is currently not the case”**
>
> In our original submission, we made a significant effort to ensure the paper is self-contained by defining all necessary concepts and even including a clean proof of the previously established LD-SQ equivalence from Brennan et al. in Appendix B. That said, we are fully committed to carefully revisiting and rereading the manuscript, with the goal of improving the clarity of our presentation, and of adding any missing elements to further enhance its self-contained nature.
>
>
> * **“Most of the text at this point barely consists of descriptions of the formulas. Central to the generalized Franz Parisi criterion is the notion of group. But the intuition behind this novelty is never fully discussed.Another important notion which is not clearly introduced is the product $\langle L_u, L_v\rangle$ “** **“I find it surprising that the Generalized Franz Parisi criterion is group dependent. I feel that this should be further addressed. “**
>
> We thank the reviewer for requesting clarification on our use of group structure in Assumption 1. To begin, we note that Assumption 1 is a deliberate relaxation: rather than requiring that $\langle L_u, L_v\rangle \geq 1$ for all pairs $u, v$, it only asks that this condition holds on average over the group orbits of $u$ and $v$ (see Section 2.1 for more details).
>
> We introduced the group-orbit version of this assumption primarily as a technical convenience in our proofs. This formulation enabled us to extend the GFP-SQ equivalence to a broader class of models. Initially, by assuming $\langle L_u, L_v\rangle \geq 1$ for all $u, v$, our equivalence applied to planted sparse models and convex truncation settings. However, we later observed that using the $Z_2$-symmetric  version still allowed the proof to go through, and crucially, extended its applicability to generalized additive models (GAMs), single-index models, and non-Gaussian component analysis (NGCA) models.
>
> To the best of our knowledge, the introduction of group symmetry in Assumption 1 currently serves as a technical tool to broaden the scope of our proof. That said, it appears to play a deep role in bridging the GFP and SQ frameworks. We therefore find this connection intriguing and believe it opens up an exciting direction for future research to understand why that is the case.
>
>
> * **"Another important notion which is not clearly introduced is the product $\langle L_u, L_v\rangle$ "**
>
> Below Definition 1, page 2 of the paper, both the inner product between functions in $L^2(Q)$ is defined and the likelihood function $ L_u$. We had a typo in Definition 1, which we will correct in the camera-ready version of our work: by writing $\langle L_u, L_v\rangle$, we meant to write $\langle L_u, L_v\rangle_Q$. Please let us know if you need any further clarification on this.
>
>
> * **“Most of the definitions involve tuples of parameters (e.g. the Generalized Franz Paris criterion is defined with respect to a triple (q, m, epsilon)) I think it would be good to recall the meaning of those parameters from time to time in the text. E.g. In lines 107 - 109, the authors give an intuition for q”**
>
> We mention the interpretation of the parameters multiple times in the paper (e.g., lines 78-86, 107-109, 120-123 in the first four pages). We will carefully read the paper and add any further remarks on this topic that we can in our camera-ready version of the paper.
>
>
>
> * **“I think the paper should be reorganized around a practical example such as the Gaussian additive model.”**
>
> We thank the reviewer for this suggestion.
>
> In our examples section, we begin with generalized additive models (GAMs) because they are among the simplest cases and have already been carefully analyzed in the prior work on the FP-criterion, particularly by Bandeira et al. (2022). There, we include an explanation of why our GFP-SQ equivalence applies in the setting of all GAMs, which reduces to a straightforward calculation, meaning that for whichever GAM one knows the SQ-hardness prediction, the GFP-hardness prediction is (essentially) the same. For added pedagogical clarity, we are happy to incorporate an earlier remark in the paper referencing a specific GAM example—such as tensor PCA—and explicitly illustrating how the GFP, FP, and SQ criteria align in their hardness predictions for that model.
>
> That said, we respectfully disagree with the suggestion to reorganize the paper entirely around such an example. Several prior works (e.g., Kunisky et al. (2019), Brennan et al. (2021), Bandeira et al. (2022)) have already provided detailed calculations of how the LD-, SQ-, and FP-criteria apply to many of these models, including GAMs. Given the space constraints of a 9-page main paper, we believe our focus should remain on the GFP criterion itself—its precise formulation, novel contributions, and broad applicability.
>
>
> * **“The theoretical definitions 2 and 3 are interesting but better intuition should be provided”**
>
> Definition 3 corresponds to the classical SDA hardness criterion, as introduced in Feldman et al. (2017) and further developed in Brennan et al. (2021). Moreover, our Definition 2 of GFP-hardness represents the main contribution of this work. We devote significant discussion to this new criterion, establish its equivalence with the SQ framework, and provide an example that highlights its separation from the FP definition proposed by Bandeira et al. (2022). Let us know if you need us to provide more details.
>
>
> * **Lines 65-67, The likelihood ratio should be properly introduced. I strongly encourage a complete introduction such as in e.g. Kunisky et al. ... I also encourage mentioning the annealed upper bound (and its connection to the FP potential) in the main paper.**
>
> We are happy to include more details of $L_u$ and its definition in the appendix of the paper. However, we respectfully disagree with the reviewer that these standard arguments---such as those thoroughly covered in the survey by Kunisky et al., or the annealed bound already explained in Bandeira et al. (2022)---belong in the main body of our paper, given the strict 9-page limit of the conference format.
>
>
> * **"The dependence on the cardinality of the group G is quite interesting and to my opinion would deserve some additional comments."**
>
> This arises as a technical consequence of certain Hölder inequality bounds used in our proofs. While we don’t think the group $G$ should be present in an optimized version of our equivalence results, notice that it is rather benign in all our applications, as in all examples $|G|$ is either $1$ or $2$.

---

> > ### Comment · Area_Chair_bohk · 2025-08-07
> >
> > Dear Reviewer Nb5E,
> >
> > Please read the author's answer and comment on which of your concerns were mitigated and which remain (if any). We have one more day left for these discussions and your participation is crucial (and mandatory).

---

### Decision · Program_Chairs · 2025-09-17

**Decision:**

Accept (oral)

**Comment:**

This paper introduces a generalized Franz–Parisi criterion and proves its equivalence with Statistical Query lower bounds under mild assumptions. This provides a rigorous bridge between physics-inspired heuristics and formal complexity theory, clarifying the computational limits of a wide class of planted models. Reviewers praised the work as a substantial conceptual advance, technically strong and broadly relevant, with several noting its originality and potential to influence both statistical physics and theoretical machine learning.

Some reviewers found the exposition dense and the positioning with respect to prior work insufficiently sharp. In rebuttal the authors committed to significant reorganization, more intuitive examples, and clarification of technical points. Reviewers acknowledged these efforts, and the consensus remained strongly positive.

Given the combination of conceptual depth, breadth of applicability, and clear relevance across communities, I view this as an outstanding contribution. Despite its demanding presentation, the work stands out in both originality and impact, and I recommend acceptance as an oral.

[SAC edit]
This is indeed a very strong contribution that tackles a relevant problem and establishes an important bridge between two methods belonging to two different domains (and communities). It is thus of wide interest (amongst the theory community) and with potential high impact. Oral presentation is recommended.